

# Evaluation of atmospheric profiles derived from single- and zero-difference excess phase processing of BeiDou System radio occultation data of the FY-3C GNOS mission

**Weihua Bai[1,2,4], Congliang Liu[1,4], Xiangguang Meng[1,4], Yueqiang Sun[1,2,4],**
**Gottfried Kirchengast[3,4,1], Qifei Du[1,4], Xianyi Wang[1,4], Guanglin Yang[5], Mi Liao[5],**
**Zhongdong Yang[5], Danyang Zhao[1,4], Junming Xia[1,4], Yuerong Cai[1,4], Lijun Liu[1,4],**
**and Dongwei Wang[1,4]**

[1] National Space Science Center, Chinese Academy of Sciences (NSSC/CAS) and Beijing
Key Laboratory of Space Environment Exploration, Beijing, China

[2] School of Astronomy and Space Science, University of Chinese Academy of Sciences,
Beijing, China

[3] Wegener Center for Climate and Global Change (WEGC) and Institute for Geophysics,
Astrophysics, and Meteorology/Institute of Physics, University of Graz, Graz, Austria

[4] Joint Laboratory on Occultations for Atmosphere and Climate (JLOAC) of NSSC/CAS,
Beijing, China, and University of Graz, Austria

[5] National Satellite Meteorological Center, Chinese Meteorological Agency, Beijing, China

*Correspondence to:* Liu Congliang (Email:liucongliang1985@gmail.com); Xiangguang Meng
(Email: xgmeng@nssc.ac.cn)

**Abstract**

The Global Navigation Satellite System (GNSS) Occultation Sounder (GNOS) is one of the
new generation payloads onboard the Chinese FengYun 3 (FY-3) series of operational
meteorological satellites for sounding the Earth's neutral atmosphere and ionosphere. GNOS
was designed for acquiring setting and rising radio occultation (RO) data by using GNSS
signals from both the Chinese BeiDou System (BDS) and the U.S. Global Positioning System
(GPS). An ultra-stable oscillator with 1-sec stability (Allan deviation) at the level of $10^{-12}$ was
installed on FY-3C GNOS, thus both zero-difference and single-difference excess phase
processing methods should be feasible for FY-3C GNOS observations. In this study we focus
on evaluating zero-difference processing of BDS RO data vs. single-difference processing, in



order to investigate the zero-difference feasibility for this new instrument, which after its
launch in September 2013 started to use BDS signals from 5 geostationary orbit (GEO)
satellites, 5 inclined geosynchronous orbit (IGSO) satellites and 4 medium earth orbit (MEO)
satellites. We used a 3-month set of GNOS BDS RO data (October to December 2013) for the

evaluation and compared atmospheric bending angle and refractivity profiles, derived from
single- and zero-difference excess phase data, against co-located profiles from ECMWF
(European Centre for Medium-Range Weather Forecasts) analyses. We also compared against
co-located refractivity profiles from radiosondes. The statistical evaluation against these
reference data shows that the results from single- and zero-difference processing are

consistent in both bias and standard deviation, clearly demonstrating the feasibility of zero-
differencing for GNOS BDS RO observations. The average bias (and standard deviation) of
the bending angle and refractivity profiles were found to be as small as about 0.05%–0.2%
(and 0.7%−1.6%) over the upper troposphere and lower stratosphere, including for the GEO,
IGSO, and MEO subsets. Zero-differencing was found to perform slightly better, as may be

expected from its lower vulnerability to noise. The validation results establish that GNOS can
provide, on top of GPS RO profiles, accurate and precise BDS RO profiles both from single-
and zero-difference processing. The GNOS observations by the series of FY-3 satellites will
thus provide important contributions to numerical weather prediction and global climate
change analysis.

**Keywords:** radio occultation, FY-3 GNOS validation, BeiDou System (BDS), excess phase,
single-differencing, zero-differencing

## 1    Introduction

The Earth's radio occultation (RO) technique (Melbourne et al., 1994; Ware et al., 1996) using
signals from the global navigation satellite system (GNSS) has been widely used to observe
the atmospheric parameters (e.g., bending angle, refractivity, temperature, pressure, and water
vapor) for applications such as numerical weather prediction (NWP) (e.g., Healy and Eyre,
2000; Kuo et al., 2000; Healy and Thepaut, 2006; Aparicio and Deblonde, 2008; Cucurull and
Derber, 2008; Poli et al., 2008; Huang et al., 2010; Le Marshall et al., 2010; Harnisch et al.,
2013) and global climate monitoring (GCM) (e.g., Steiner et al., 2001, 2009, 2011, 2013;



Schmidt et al., 2005, 2008, 2010; Loescher and Kirchengast, 2008; Ho et al., 2009, 2012; Foelsche et al., 2011a; Lackner et al., 2011).

The RO concept was experimentally tested by the first experimental Global Positioning System/Meteorology (GPS/MET) mission launched in 1995 right after the full operational capacity of GPS was achieved (Ware et al., 1996; Kursinski et al., 1996; Kuo et al., 1998). GPS/MET has demonstrated the unique properties of the GPS RO technique, such as high vertical resolution, high accuracy, all-weather capability and global coverage (Ware et al., 1996; Gorbunov et al., 1996; Rocken et al., 1997; Leroy, 1997; Steiner et al., 1999).

The subsequent LEO satellite missions such as the CHAllenging Minisatellite Payload (CHAMP) (Wickert et al., 2001, 2002), the Constellation Observing System for Meteorology, Ionosphere and Climate (COSMIC) (Anthes et al., 2000, 2008; Schreiner et al., 2007), the Gravity Recovery And Climate Experiment (GRACE) (Beyerle et al., 2005; Wickert et al., 2005), and the Meteorological Operational (MetOp) satellites (Edwards and Pawlak, 2000; Luntama et al., 2008) satellites have further affirmed the long term stability and remarkable consistency (e.g., <0.2 – 0.5 K in temperature) of RO observations from different RO missions (Foelsche et al., 2009, 2011a).

The development of GNSS such as China's BeiDou Navigation Satellite System (BDS), Russia's GLObal NAvigation Satellite System (GLONASS), and the European Galileo system, has significantly enhanced the availability and capacity of the GPS-like satellites which will make RO even more attractive in the future. These new GNSS navigation satellites together with planned LEO missions will offer much more RO observations. One of these LEO missions is China's GNss Occultation Sounder (GNOS) onboard first time on the FengYun 3 series C satellite (FY-3C), which was successfully launched on 23 September 2013 (Liao et al., 2016).

FY-3C GNOS, developed by National Space Science Center/Chinese Academy of Sciences (NSSC/CAS), is the first BDS/GPS compatible sounder and combines a state-of-the-art RO receiver with an ultra-stable oscillator. The future satellites of the Chinese FY-3 series of operational meteorological satellites, with the launch of FY-3D upcoming later in 2017, will as well have GNOS on board, similar to the MetOp series of European satellites with its GRAS instruments (Loiselet et al., 2000).



The GNOS instrument consists of three antennas, three radio frequency (RF) units and a data processor (Figure 1a), which uses high-dynamic, high-sensitivity signal acquisition and tracking techniques, as well as an antenna with stable phase center. Additionally, the different features of BDS and GPS signals have been taken into account in GNOS design. GNOS can

observe the atmosphere and ionosphere and its detection height range is from Earth surface to around 800 km altitude. So far, a large dataset of FY-3 GNOS RO observations has been obtained. Figure 1b illustrates the number of both the GPS RO and BDS RO events processed over the three months from October to December 2013, which are used for the single- and zero-difference excess phase analysis in this paper.

Regarding the excess phase processing, a single-difference method removes the LEO satellite clock offset by the difference between the GNSS occultation satellite and its GNSS reference satellite (Wickert et al., 2002). Comparing with the original double-difference method (Ware et al., 1996; Rocken et al., 1997), it uses the solved GNSS satellite clock offset estimates instead of further differencing between the GNSS satellites and a GNSS ground station, hence

single-difference method can minimize the effects of ground data error sources during the GPS clock offset estimation process (Hajj et al., 2002; Schreiner et al., 2010). Because single-differencing needs no ground station data its data processing is simpler and easier to realize. Therefore the single difference approach has become widely used in RO data processing after the switch-off of the GPS "selective availability" (SA) mode as of May 2000 (Hajj et al.,

2002), which made GPS clock offset estimation sufficiently reliable.

Even more recently, zero-difference processing started to be used (Beyerle et al. 2005; Wickert et al., 2005), which can compute excess phase data by applying prior estimated LEO and GNSS clock offsets without need of a reference satellite or ground station. However, it requires the LEO receiver to be clocked by an ultra-stable oscillator that was so far available

only for the GRACE and MetOp missions (Beyerle et al. 2005; Luntama et al., 2008). The FY-3 GNOS instrument is clocked by such an ultra-stable oscillator as well.

BDS is China's global navigation satellite system designed to provide global coverage around 2020, with positioning, navigation, timing, and short-message communication service capabilities (Li, 2016). So far, BDS can provide a good regional coverage in the Asia-Pacific

area with an incomplete constellation. For the time period of this study in fall 2013, the FY-3C GNOS satellite received 5 geostationary orbit (GEO) satellites, 5 inclined geosynchronous



orbit (IGSO) satellites and 4 medium earth orbit (MEO) orbit satellites signals to conduct the radio occultation.

This still growing constellation also provides a practical motivation for zero-differencing because not all of the FY-3C GNOS BDS RO events can be processed by single-differencing, since the incomplete BDS system cannot provide reference satellites for all RO events. On the other hand, the ultra-stable oscillator driving the GNOS receiver makes zero-differencing anyway attractive to be potentially used as the method of choice for all BDS RO events. To investigate the feasibility of the zero-difference algorithm for BDS RO data processing, and to evaluate the quality of the retrieved RO data products, we therefore perform in this study a comparative analysis of zero- and single-difference processing for GNOS.

The paper is structured as follows. Section 2 provides a description of our single- and zero-difference excess phase processing. Section 3 presents the FY-3C GNOS datasets and the methods for the inter-comparison analysis. Section 4 presents the statistical analysis results for the various reference datasets. Finally, conclusions have been drawn in Section 5.

## 2   Calculation of the FY-3C GNOS excess phase profiles

Excess phase is a key variable during the radio occultation data processing and GNSS satellite and LEO satellite clock errors are main factors effecting the excess phase accuracy. As summarized above, these two clock error components can either be eliminated by double-differencing, or (for GPS after the SA mode has been deactivated) the GNSS clock errors are estimated and subtracted and so single-differencing can be applied, or (given an ultra-stable oscillator at the LEO) both clock errors are estimated and subtracted and so zero-differencing is possible. Recently, for its higher complexity and degraded accuracy, double-differencing is rarely used. In this section we describe the single- and zero-difference procedures, which we used for the FY-3C GNOS excess phase processing.

### 2.1   Basic algorithm of the excess phase processing

The GNOS RO excess phase processing determines the total excess phase, which is caused by both the atmosphere and ionosphere, of the GPS L1, L2 and BDS B1, B2 signals as a function of coordinate (GPS) time in the Earth-Centered Inertial (ECI) True of Date (TOD) reference



frame. The inputs to process are GPS, BDS and LEO satellite positions, velocities and clock offsets as a function of coordinate time, LEO satellite attitude information, carrier phase measurements, antenna phase center information, and Earth orientation information.

The outputs of this process include GPS time of the RO event observations, where we adopt the LEO's signal reception time, GPS L1, L2 and BDS B1, B2 total excess phases, position and velocity of the LEO satellite at signal reception time, and position and velocity of the GNSS satellite at signal transmission time. Hereafter, we will use the term GNSS to refer to GPS and BDS satellites, as well as use $L$ to denote the excess phases not only for GPS signals but also for BDS signals. Specifically, in this study, we use the BDS satellite data as orbital data inputs and outputs, while time-wise also using GPS time for the processing of the BDS data. Figure 2 illustrates the geometrical basis of the differencing procedures as part of the excess phase processing.

The observed carrier phase $L_{a,i}^{b}$ (in units of meter) at frequency $i$ between the LEO receiver satellite $A$ ($a$) and the occulting GNSS transmitter satellite $B$ ($b$), as shown in Figure 2, is the essential raw observable which is modeled as

$$
\begin{aligned}
L_{a,i}^{b}(t_r) = & \rho_a^b(t_r) + c\left(\delta t_a(t_r) - \delta t_{a,rel}(t_r)\right) - c\left(\delta t^b\left(t_r - \tau_a^b\right) - \delta t_{rel}^b\left(t_r - \tau_a^b\right)\right) + \delta\rho_{a,rel}^b(t_r) \\
& + \Delta\phi_{a,i}^b(t_r) + \delta\rho_{a,ion,i}^b(t_r) + \delta\rho_{a,trop,i}^b(t_r) \, ,
\end{aligned}
\tag{1}
$$

where $t_r$ is receive time, $c$ speed of light in vacuum (m/s), $\rho_a^b$ geometric range between $a$ and $b$ at receive time, $\delta t_a$, $\delta t^b$ offsets between receive time and proper time and transmit time and proper time, respectively, $\delta t_{a,rel}$, $\delta t_{rel}^b$ offsets between proper time and coordinate time due to special and general relativity for receiver clock and GNSS satellite clock, respectively, $\tau_a^b$ light travel time between receiver and transmitter in vacuum, $\delta\rho_{a,rel}^b$ gravitational delay between receiver and transmitter, $\Delta\phi_{a,i}^b$ phase wind-up correction at receive time, $\delta\rho_{a,ion,i}^b$ ionospheric excess phase between receiver and transmitter satellite, and $\delta\rho_{a,atm,i}^b$ neutral atmospheric excess phase between receiver and transmitter satellite. The ionospheric and neutral atmospheric components $\delta\rho_{a,atm,i}^b$ and $\delta\rho_{a,ion,i}^b$ jointly are the desired total excess phase to be determined based on Eq. (1).





Needed for single-difference processing only, the carrier phase observable $L_{a,i}^c$ at frequency $i$ between LEO receiver $A$ ($a$) and reference GNSS satellite $C$ ($c$) is formally very similar to the one of the occultation link A-B and modeled as

$$L_{a,i}^c(t_r) = \rho_a^c(t_r) + c\left(\delta t_a(t_r) - \delta t_{a,rel}(t_r)\right) - c\left(\delta t^c\left(t_r - \tau_a^c\right) - \delta t_{rel}^c\left(t_r - \tau_a^c\right)\right) + \delta\rho_{a,rel}^c(t_r)$$
$$+ \Delta\phi_{a,i}^c(t_r) + \delta\rho_{a,ion,i}^b(t_r) \,,$$

(2)

where the superscript $c$ denotes the reference GNSS satellite and the meaning of the terms is otherwise as for Eq. (1). Since the reference link A-C crosses only (a part of) the ionosphere, the atmospheric excess phase term does not appear in Eq. (2).

The geometric range $\rho_a^{b,c}$, of the occultation link A-B or the reference link A-C, can be computed by

$$\rho_a^{a,c} = \sqrt{\left(X^{b,c} - X_a\right) + \left(Y^{b,c} - Y_a\right) + \left(Z^{b,c} - Z_a\right)} \,,$$

(3)

where ($X_a$ $Y_a$ $Z_a$) denotes the coordinates of the LEO satellite ($a$) at receive time and ($X^{b,c}$ $Y^{b,c}$ $Z^{b,c}$) denotes the coordinates of the GNSS satellite $b$ or $c$ at transmit time.

The GNSS satellite orbits (positions and velocities) and the GNSS clock offset estimates $\delta t^{b,c}$ are provided by the International GNSS Service (IGS) and applied as needed. Using the orbit information, the periodic relativistic effect of the GNSS satellite clock $\delta t_{rel}^{b,c}$ can be modeled by

$$\delta t_{rel}^{b,c} = -2\frac{\overline{r^{b,c}} \cdot \overline{v^{b,c}}}{c^2} \,,$$

(4)

where $r^{b,c}$ and $v^{b,c}$ are the GPS satellite position and velocity vectors at signal transmit time.

The gravitational delay $\delta\rho_{a,rel}^{b,c}$ can be modeled by

$$\delta\rho_{r,rel}^{b,c} = \frac{2GM_E}{c^2}\ln\left(\frac{r^{b,c} + r_r + \rho_r^{b,c}}{r^{b,c} + r_r - \rho_r^{b,c}}\right) \,,$$

(5)

where $G$ is Newton's gravitational constant, $M_E$ is the Earth's mass, and $r^{b,c}$ and $r_a$ are the transmitter and receiver radial positions at signal transmit and receive times, respectively.



The phase wind-up correction term $\Delta\phi_{a,i}^{b,c}$ can be modeled in the form

$$\Delta\phi_{a,i}^{b,c} = \text{sign}\left(\hat{k}\cdot\left(\overline{D}\times D\right)\right)\cdot\cos^{-1}\left(D\cdot\overline{D}\big/\left(|D|\cdot\left|\overline{D}\right|\right)\right),\tag{6}$$

where $\hat{k}$ is the unit vector from transmitter to receiver and $D$ and $\overline{D}$ are so-called effective dipole vectors; for details on this modeling see Kouba (2015).

## 2.2 Single-difference processing

In the single-difference processing we use Eq. (1) as basic equation and adopt Eq. (2) as auxiliary equation. GNSS clock offsets are subtracted and Eqs. (3) to (6) are applied to model and subtract also the GNSS-related geometric and relativistic terms from the occultation and reference link so that only the excess phases and LEO clock offsets remain.

Next, employing Eq. (2), the excess phase of the reference link (which is only an ionospheric excess phase $\delta\wp_{a,ion,i}^{c}$) can be effectively eliminated by the classical dual-frequency ionospheric correction of L1 and L2 phases (e.g., Ware et al. 1996). That is, an ionosphere-corrected phase $L_{a,3}^{c}$ can be calculated for the reference link by

$$L_{a,3}^{c} = L_{a,1}^{c}(t_r) + c_2\left\langle L_{c,1}^{a}(t_r) - L_{c,2}^{a}(t_r)\right\rangle\tag{7}$$

or

$$L_{a,3}^{c} = L_{a,2}^{c}(t_r) + c_1\left\langle L_{c,1}^{a}(t_r) - L_{c,2}^{a}(t_r)\right\rangle,\tag{8}$$

where $\langle\cdot\rangle$ denotes moving-average smoothing (over 2 seconds) and where $c_1 = f_1^2/(f_1^2 - f_2^2)$ and $c_2 = f_2^2/(f_1^2 - f_2^2)$. $c_1$ and $c_2$ are just constants in which $f_1$ and $f_2$ are the frequencies of the L1 and L2 signals, respectively.

Finally, the effects of the receiver clock, $c\left(\delta t_a(t_r) - \delta t_{a,rel}(t_r)\right)$, are eliminated by single-differencing (SD), that is by the subtraction of the reference-link phase $L_{a,3}^{c}$ from the occultation-link phases $L_{a,i}^{b}$, so that we obtain the desired SD-based total excess phase $\Delta L_{a,i}^{SD}$,

$$\Delta L_{a,i}^{SD}(t_r) = L_{a,i}^{b}(t_r) - L_{a,3}^{c}(t_r) = \delta\wp_{a,ion,i}^{b}(t_r) + \delta\wp_{a,atm,i}^{b}(t_r).\tag{9}$$



## 2.3 Zero-difference processing

The single-difference approach has some advantages comparing with double-difference, as noted in the introduction above, and has therefore been widely used in GPS RO data processing. However, it is difficult to find a suitable reference satellite for each RO event to
calculate the excess phase using single-difference when the GNSS section is still an incomplete constellation such as the current BDS. Zero-difference processing also will likely produce lower-noise excess phase data than single-differencing, from applying the estimated LEO clock offsets and avoiding the use of a reference link (being an additional error source), if the LEO receiver is equipped with an ultra-stable oscillator such as in case of the GNOS
instrument.

In the zero-differencing (ZD) approach we just employ Eq. (1) directly and model and subtract all relevant terms as summarized in subsection 2.1 above, including the GPS and LEO clock offsets, so that we obtain the desired ZD-based total excess phase $\Delta L_{a,i}^{ZD}$,

$$
\begin{aligned}
\Delta L_{a,i}^{ZD}(t_r) &= \delta \wp_{a,ion,i}^{b}(t_r) + \delta \wp_{a,trop,i}^{b}(t_r) = \\
&L_{a,i}^{b}(t_r) - \left( \rho_a^b(t_r) + c\left( \delta t_a(t_r) - \delta t_{a,rel}(t_r) \right) - c\left( \delta t^b\left(t_r - \tau_a^b\right) - \delta t_{rel}^b\left(t_r - \tau_a^b\right) \right) + \delta \wp_{a,rel}^{b}(t_r) + \Delta \phi_{a,i}^{b}(t_r) \right).
\end{aligned}
\tag{10}
$$

## 3  Differencing and analysis methods for the GNOS BDS RO data

### 3.1  Necessity of zero-differencing for GNOS BDS RO data

As aforementioned, the single-difference approach involves a GNSS reference satellite, which should have high signal to noise ratio (SNR) and high phase measurement accuracy. In order
to use that specific reference satellite that most likely has the best signal quality and lowest ionospheric influence, our FY-3C GNOS processing chooses the GNSS satellite with highest elevation angle as the reference satellite.

The largest gain and half-power beam width of GNOS's POD antenna is 5 dB and 40 degree, respectively, and the normal vector of the antenna plane points to the zenith, hence the
antenna gain increases with increasing elevation angle. Therefore, ignoring the multi-path effect, the positioning channel carrier phase error increases with decreasing elevation and, ultimately, the satellite tracking will lose the lock when the elevation angle becomes very small.



Figure 3 illustrates the GNOS in-orbit testing results of the BDS B1 and B2 carrier phase observation error standard deviation, as a function of elevation angle. As can be clearly seen, both the B1 and B2 carrier phase measurement errors decrease with increasing elevation angle. At elevation angles larger than 10 deg, the B1 and B2 carrier phase errors are less than 2 mm.

Therefore, currently we select the reference satellite for the single-difference method from those satellites whose elevation angle is at least larger than 10 deg.

Applying this 10-deg elevation threshold criterion, we counted the numbers of GNOS BDS RO events with and without reference satellites. In this statistical analysis all the GNOS BDS RO events that occurred from 1 Oct 2013 to 31 Dec 2013 were included. Figure 4 shows that

during these 3 months there were 13564 GNOS BDS RO events in total, of which about 16% had a maximum elevation angle of possible reference satellites below 0 deg, and a total of 20% had their reference satellites below 10 deg. In practice, less than 0 deg means that there is in fact no reference satellite in view and less than 10 deg still implies that the reference satellites' tracking accuracy is considered not sufficient for the single-differencing. Therefore,

these 20% of BDS RO events can meaningfully be processed only by the zero-difference approach, since the still sub-global BDS system coverage cannot satisfy the 10-deg elevation threshold criterion for these events.

### 3.2   GNOS BDS RO data and statistical analysis method

To evaluate the performance of the zero- and single-difference methods, we have conducted a comparison analysis of the retrieved FY-3C GNOS BDS RO bending angle and refractivity data for the selected 92 days from 1 Oct 2013 to 31 Dec 2013, retrieved by either including the single-difference or zero-difference method in the excess phase processing.

In our data processing, a quality control algorithm has been used. The processing statistics we

obtain show that, after quality control, the number of RO events obtained by zero-differencing is higher by about 13% than the one obtained by single-differencing, which we find is due to some ineffective reference BDS satellite links during the single-difference processing. The geographic and local time distribution of the RO events that also have proper BDS reference satellites for single-difference processing is shown in Figure 5.



Figure 5a shows that the geographic distribution of events well reflects the different BDS orbit types. BDS-GEO RO events mainly distribute in the southern and northern hemisphere high latitude zones along the longitude sector of the Chinese region. The number of BDS-IGSO RO events is highest, almost equal to the number of GEO and MEO RO events together. The BDS-IGSO RO event coverage forms a quasi-global "8" shape, with the larger oval over the American, Pacific, and Atlantic Ocean areas, and the somewhat smaller oval over southeast Asia, northwest Australia, Pacific, and Indian Ocean areas. Similar to the typical distribution of GPS RO events (e.g., Pirscher et al., 2007; Anthes et al., 2008), the BDS-MEO RO events show essentially global coverage, with more RO events in the middle and high latitude zones and less at low latitudes.

Figures 5b and 5c show the distribution of the RO events in a complementary way with focus on local time, again reflecting well the different BDS orbit types and their impact on RO event locations in space and time. It can be seen that the BDS-GEO RO events occur during all 24 hours of the day, while the BDS-IGSO and BDS-MEO RO events distribute mainly in the 9:00-11:00h and 21:00-23:00h local-time ranges (best seen in Fig. 5c). In particular at low and middle latitudes, equatorward of about 50° to 60°, no BDS RO events at all occur within about 00:00-08:00h and 12:00-20:00h local time (see Fig. 5b). This is due to the near-polar sun-synchronous orbit of the FY-3C meteorological satellite, similar to the European MetOp satellites as analyzed by Pirscher et al. (2007).

The distribution of the GNOS BDS RO events processed by using zero-differencing (not separately shown) is very similar to Figure 5, though with slightly more RO events (2623 BDS-GEO, 4820 BDS-IGSO, and 2863 BDS-MEO) that had passed the quality control.

For producing the statistical analysis results compared to reference data, we calculated the fractional error of the retrieved bending angle (*BA*) and refractivity (*R*) profiles in the form,

$$E_{BA} = 100 \times \frac{BA - BA_{ref}}{BA_{ref}} \ [\%] \ , \tag{11}$$

$$E_R = 100 \times \frac{R - R_{ref}}{R_{ref}} \ [\%] \ , \tag{12}$$

where *E* denotes the estimated fractional error profiles (against the reference data) for which ensemble estimates of biases and standard deviations are illustrated in the result figures.





As reference data we used analysis data from the European Centre for Medium-Range Weather Forecasts (ECMWF) as well as radiosonde data obtained from the global radiosonde archive of the National Oceanic and Atmospheric Administration–National Centers for Environmental Information (NOAA-NCEI).

The ECMWF analysis data were used as 6-hourly fields (00, 06, 12, 18 UTC time layers every day) at a horizontal resolution of about 300 km and with 137 vertical model levels (yielding about 0.5 km to 1.5 km resolution over the upper troposphere/lower stratosphere domain of interest). Vertical profiles co-located with the GNOS RO profiles were extracted from these fields by bi-linear interpolation in latitude and longitude to the mean RO event

location, using the nearest-neighbor time layer of the RO event time. Since the GNOS data were not assimilated into the ECWMF system the data are clearly independent. The radiosonde profiles were about 0.5 km to 3 km vertical resolution over the domain of interest, and were used with a ±1 deg lat-lon / ±1 hour collocation criterion to the RO event.

## 4    GNOS BDS RO single-difference and zero-difference results analysis

The target domain for the comparative statistical analysis is from 5 km to 35 km height (upper troposphere and lower stratosphere, UTLS), since commonly the data quality above 35 km and below 5 km is less good, due to the ionospheric effects and tropospheric multipath effects, respectively (e.g., Scherllin-Pirscher et al., 2011a, 2011b; Steiner et al., 2013). We first inspect difference statistics to ECMWF and subsequently to radiosondes.

### 20   4.1    Comparison analysis of bending angle with ECMWF data

Figure 6 shows the statistics of the GNOS BDS RO bending angle results, for the different BDS subsystems (GEO, IGSO, MEO) and the full BDS (Total), for both single-differencing (Fig. 6a) and zero-differencing (Fig. 6b). The Bias and StdDev profiles have been calculated from the large ensembles of these event datasets, based on the fractional difference profiles

according to Eq. (11).

In line with expectations, the biases and standard deviations are slightly smaller for the zero-differencing than for the single-differencing (though more standard deviation suppression might be expected from avoiding the reference link computation; e.g., Schreiner et al., 2009) but in general they are very similar. Both cases show a small negative bias of around –0.15%





against ECMWF, and a standard deviation of around 1.5%. At least part of the bias is likely from slight differences in vertical geolocation of GNOS and reference profiles, for which ensuring rigorous consistency is a subtle process (Scherrlin-Pirscher et al., 2017). Likewise, part of the standard deviation is from representativeness error between the GNOS and

ECWMF profiles, since even though being co-located in mean location they have different detailed locations and resolutions (Foelsche et al., 2011b; Scherllin-Pirscher et al., 2011b).

Overall the results confirm a high quality of the GNOS retrievals, in line with recent results by Liao et al. (2016), and a robust zero-difference processing being a viable alternative for the single-difference processing. The results also show, that the BDS retrievals achieve a quality

comparable to what is well established for GPS retrievals ( ). Thanks to the diversity of BDS orbits, we can also demonstrate for the first time RO retrievals from occultations with GNSS transmitters not in medium Earth orbit (MEO). The results clearly indicate that also the GNSS transmitters in GEO and IGSO can provide a quality comparable to the ones in MEO.

## 4.2   Comparison analysis of refractivity with ECMWF data

Figure 7 shows the statistics of the GNOS BDS RO refractivity results, again for the different BDS subsystems (GEO, IGSO, MEO) and the full BDS (Total), for both single-differencing (Fig. 7a) and zero-differencing (Fig. 7b). The Bias and StdDev profiles have been calculated from these large BDS event ensembles based on the fractional refractivity difference profiles according to Eq. (12).

Similar to the bending angle results (Figure 6), the biases and standard deviations for the refractivity results are a bit smaller for the zero-differencing than for the single-differencing but are otherwise quite similar. Both cases show a small negative bias of around –0.05% against ECMWF, and a standard deviation of near 0.8% (single-differencing more near 0.9%). This reduction of bias and standard deviation magnitudes compared to the bending angle (by

about a factor of two) is due to the filtering properties of the Abelian integral that transforms the bending angle to refractivity profiles (Rieder and Kirchengast, 2001; Scherllin-Pirscher et al., 2011a; Schwarz et al., 2017).



Overall the refractivity results confirm the messages summarized in Sect. 4.1 based on the bending angle results. That is, they underline the high quality of the GNOS BDS retrievals as being comparable to GPS retrievals, the robustness of both the zero- and single-difference processing, and the reliable retrieval quality also for the RO events with GNSS transmitters

not on MEO satellites but rather on GEO and IGSO satellites.

### 4.3  Comparison analysis of refractivity with radiosonde data

Figure 8 shows the single- and zero-difference results for refractivity statistics, again Bias and StdDev profiles, here against collocated radiosonde profiles and only for the whole set of BDS RO events, since the number of collocations is more limited. The number of RO events

entering into the statistics is also strongly height-dependent in this case and is therefore shown not only as (maximum) number in the legend but also as height profiles in a side panel (Figure 8, right).

Given the smaller ensemble size of about 50 to 200 events (depending on height) and the less strict collocation, and thus somewhat higher representativeness error than for the ECMWF

data extracted at the mean RO event location, these refractivity results are expected to exhibit somewhat more deviations than those in Figure 7. As Figure 8 shows, the bias is nevertheless still fairly small, near –0.3%, and the standard deviation is near 0.95%, i.e, still below 1%.

In summary, also the comparison to this entirely independent radiosonde dataset underpins the finding that both the zero- and single-differencing do a robust job and that the GNOS BDS

retrievals exhibit a high performance similar to GPS retrievals that have been established earlier to compare well to quality radiosondes (Anthes, 2011; Ladstaedter et al., 2015).

### 5  Conclusions

In this study we have introduced our single- and zero-difference excess phase processing of BeiDou System (BDS) RO data of the FY-3C GNOS mission and evaluated the quality of

atmospheric profiles derived based on this single- and zero-difference processing.

Single-differencing does not need to correct the receiver clock offset, thus it has lower requirements on the receiver clock stability. However, it requires a proper reference GNSS satellite and will induce some of this reference satellite's positioning and carrier phase measurement errors into the RO processing. The advantage of the zero-difference algorithm is



its independence from reference satellites, but it requires a receiver clock of very high quality (ultra-stable oscillator such as available for GNOS) to obtain a highly accurate receiver clock offset estimate, which nevertheless can leave some residual errors after the clock offset correction.

Because BDS currently still is a sub-global navigation system, we found that about 20% of the GNOS BDS RO events do not have proper reference satellites for single-differencing, providing another argument for a zero-difference alternative. We performed a comparative analysis of the zero-difference and single-difference excess phase processing chains for the FY-3C GNOS BDS RO observations, in which independent reanalysis data from ECWMF

and collocated high-quality data from radiosondes have been used as reference for evaluating the retrieved bending angle and refractivity profiles over the upper troposphere and lower stratosphere (UTLS, 5 km to 35 km).

The results showed that the GNOS BDS RO profiles derived by using both the zero-difference and single-difference algorithms exhibit very good consistency with the ECMWF

and radiosonde data. The zero-difference method appeared to perform slightly better than the single-difference method, especially visible at stratospheric altitudes (above 15 km).

Comparing to ECMWF data, the average UTLS bending angle bias was found near –0.15% and the associated average standard deviation near 1.5%; the average refractivity bias was accordingly found as small as around –0.05% and the associated standard deviation at about

0.8%. Comparing to radiosonde data, the GNOS BDS RO refractivity profiles both from zero- and single-difference processing also showed high consistency, with the average refractivity bias in the UTLS found near –0.3% and the associated standard deviation near 0.95%, i.e., also below 1%, despite increased representativeness error in this latter comparison.

Overall these results indicate high quality of the GNOS BDS retrievals, and a robust zero-

difference processing that is a viable alternative for the single-difference processing. The results also showed, that the BDS retrievals achieve a quality comparable to the established GPS retrievals. Based on the diversity of BDS orbits, we also demonstrated for the first time RO retrievals from occultations with GNSS transmitters not in medium Earth orbit (MEO). We found that also the GNSS transmitters in geostationary Earth orbit (GEO) and in inclined

geo-synchronous orbit (IGSO) provide a quality comparable to the ones at MEO satellites.





Currently, the GNSS Receiver for Atmospheric Sounding (GRAS) onboard the European meteorological satellite series MetOp and the GNOS occultation receiver onboard the Chinese meteorological satellite series FY-3 are the two RO instruments for long-term operational observations that include an ultra-stable crystal oscillator featuring a very high-quality Allen

5 variance at the $10^{-12}$ second accuracy level. In the future, additional RO missions such as COSMIC-2, MetOp-SG, and advanced-GNOS instruments will expand on this high-quality basis. For these operational backbone missions, leading the field with their data quality, the zero-difference method will generally perform better and will thus likely replace the single-difference method in the future.

10 As established with the initial evaluation in this study, the GNOS observations by the series of FY-3 satellites may well provide an essential contribution in this context of the leading instruments, for the benefit of numerical weather prediction and climate change analysis.

### Acknowledgements

15 This research was supported by the National Natural Science Foundation of China (Grant Nos. 41505030, 41405039, 41405040 and 41606206) and by the Feng Yun 3 (FY-3) Global Navigation Satellite System Occultation Sounder (GNOS) development and manufacture project led by NSSC/CAS. The research at WEGC was supported by the Austrian Aeronautics and Space Agency of the Austrian Research Promotion Agency (FFG-ALR) under projects

20 OPSCLIMPROP (Grant No. 840070) and OPSCLIMTRACE (Grant No. 844395). The ECMWF (Reading, UK) is thanked for access to their archived analysis and forecast data (available at http://www.ecmwf.int/en/forecasts/datasets) and NOAA-NCEI (Boulder, CO, USA) for access to their radiosonde data archive (available at http://www.ncdc.noaa.gov/data-access/weather-balloon-data). The software code used for this study does not belong to the

25 public domain and cannot be distributed. To access the relevant result files of this study, please contact the corresponding author.

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



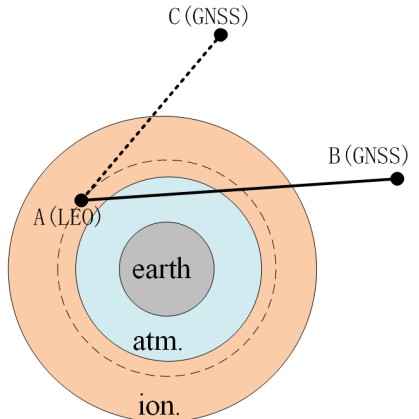

**Figure 2.** Schematic geometry of GNSS radio occultation for single-differencing (using link A-C in addition to link A-B) and zero-differencing (using link A-B only).

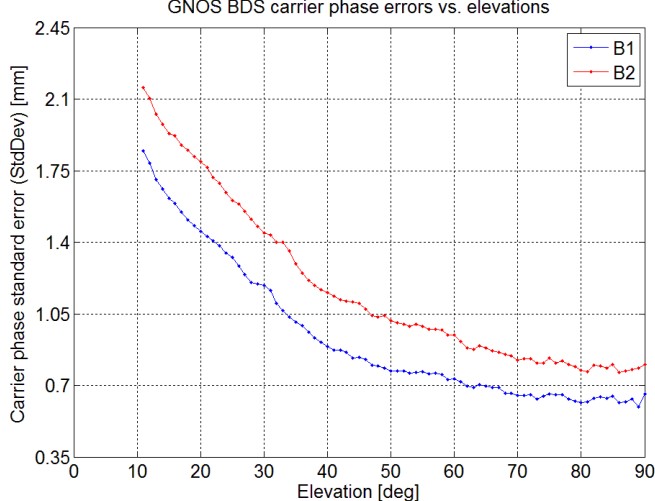

**Figure 3**. Statistics of FY-3C GNOS BDS carrier phase standard deviations (blue, B1 signal carrier phase; red, B2 signal carrier phase) as function of elevation angle, calculated by using positioning channel measurements.



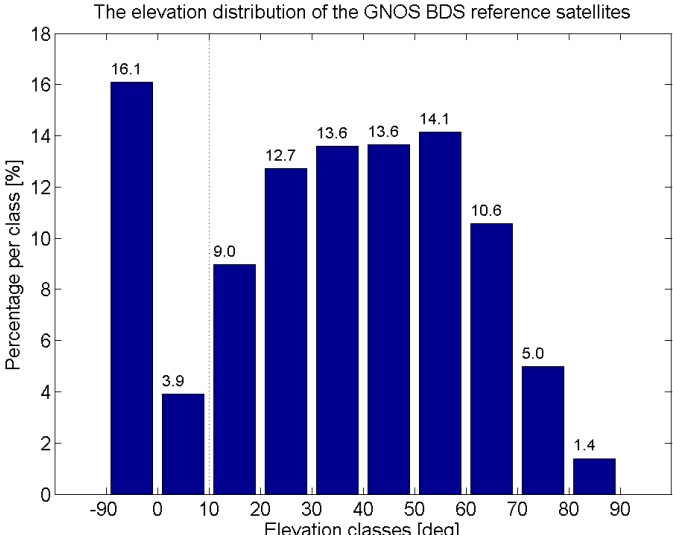

**Figure 4**. Histogram of the maximum elevation angle of the BDS reference satellites, with the statistics based on the 13564 BDS RO events that occurred over October-December 2013.



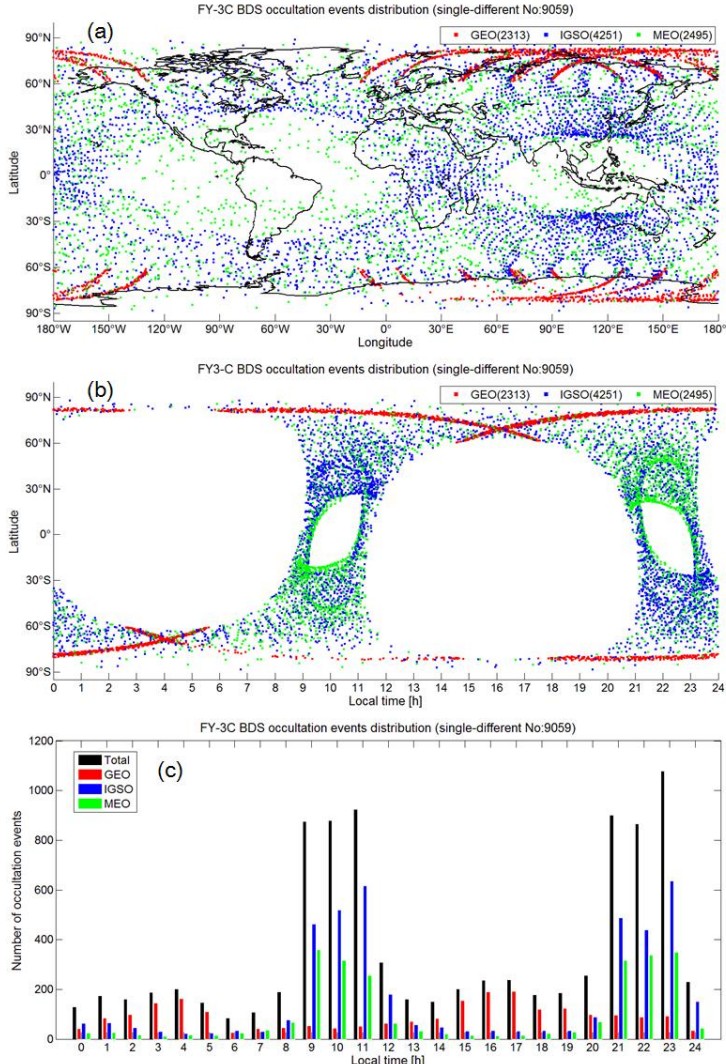

**Figure 5**. Geographical and local time distribution of the GNOS BDS RO events that have proper BDS reference satellites for single-difference processing (red, from the BDS-GEO satellites; blue, from BDS-IGSO; green, from BDS-MEO; numbers in parentheses denote the associated number of events during Oct-Dec 2013). Distributions are shown as function of latitude and longitude **(a)**, as function of local time and latitude **(b)**, and in histogram-style as function of local time (black herein denotes from all BDS satellites) **(c)**.



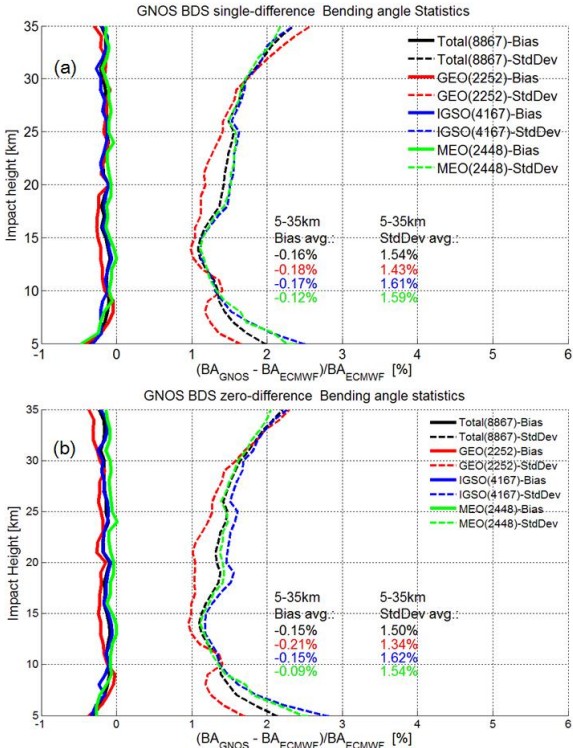

**Figure 6**. Mean difference (Bias) and standard deviation (StdDev) statistics of the GNOS bending angle profiles retrieved by using excess phases from the single-difference processing **(a)** and the zero-difference processing **(b)**, respectively, with the co-located ECMWF bending angle profiles used as reference. Bias (solid) and StdDev (dashed) profiles are shown for the set of all BDS RO events (black), and the subsets of BDS-GEO (red), BDS-IGSO (blue), and BDS-MEO (green). Legends also indicate the numbers of events involved and the average Bias and StdDev values over the 5–35 km range.





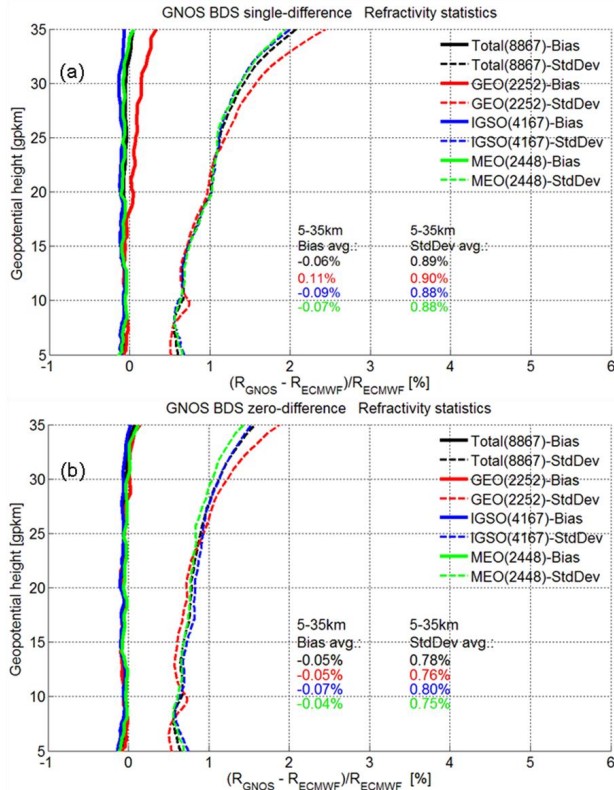

**Figure 7**. Mean difference (Bias) and standard deviation (StdDev) statistics of the GNOS refractivity profiles retrieved by using excess phases from the single-difference processing **(a)** and the zero-difference processing **(b)**, respectively, with the co-located ECMWF refractivity profiles used as reference. Same layout as Figure 6; see that caption for further description.



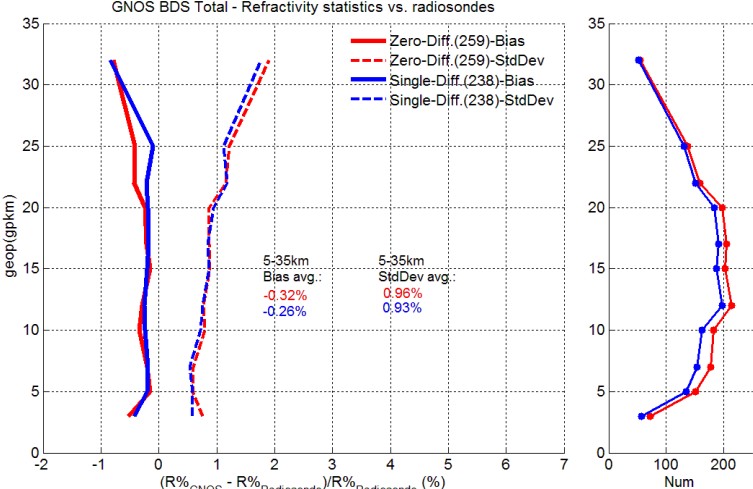

**Figure 8**. Mean difference (Bias) and standard deviation (StdDev) statistics of the GNOS refractivity profiles retrieved by using either zero-differencing (red) or single-differencing (blue), with collocated radiosonde refractivity profiles used as reference (±1 lat-lon / ±1 h collocation criterion). Bias (solid) and StdDev (dashed) profiles as well as the number-of-event profiles (small right-hand-side panel) are shown for the total set of BDS RO events. The legends also indicate the numbers of events involved and the average Bias and StdDev values within 5–35 km.