# Peer review of "Evaluation of atmospheric profiles derived from single- and"

_Atmospheric Measurement Techniques, 2017_

## Referee Comment (RC1) · Anonymous Referee #1 · 7 Jul 2017

**1   General Comments**

This article presents atmospheric profiling results derived from radio occultation of satellites in the BeiDou constellation over three months, as processed by single- and zero- differencing algorithms. The derived bending angles and refractivity profiles are compared to results from the ECMWF and radiosondes collocated geospatially and temporally with the profiles from the GNOS instrument.

[Figure]

This paper is well organized and does a good job of describing the processing methodologies. The resulting BDS profiles are fairly consistent with both other radio occultation measurement results from the ECMWF and localized radiosondes. The results are encouraging in both the use of ultra-stable oscillators for radio occultation collection instruments (zero-differencing), and the use of the BeiDou signals as remote sensing sources for future atmospheric sensing satellite missions. I only have a few specific comments and suggestions that I'd like to see further expanded upon in the revision.

**2  Specific Comments**

The authors mention, early in the paper, that the GNOS receiver is capable of collecting both GPS and BDS data. However, I am slightly confused as to whether any GPS data were used in your single/zero-differencing studies. You make a distinction on Page 6 that the term "GNSS" refers to both GPS and BDS satellites, but it seems like only BDS satellites are used for the occultation measurements, and perhaps GPS is just used for timing? It would be interesting to the reader to compare occultation results from your same algorithms, but with GPS data over the same time and spatial intervals.

Another related point, I am curious as to why your results are negatively biased from both the ECMWF and radiosondes. The authors make a comment as to the differences in the vertical geolocations of the profiles in comparison to the reference data, but it is odd that all the different types of BDS satellites (GEO, IGSO, MEO) are negatively biased. Again, if the authors were to process GPS data from the same times/locations with their single/zero-differencing algorithms, it could be another way to validate their methodologies and results.

The authors use radiosonde measurements within a +/- 1 deg lat-lon/ +/- 1 hour collocation criterion to validate an RO event for part of their analysis. This range can be on the order of a 200 km x 200 km box, over the course of an hour. Do the authors have an explanation or reference to the stability of the atmosphere over these spatial and time ranges?

**3  Technical Corrections**

- Page 3, lines 13-14: the word "satellites" is repeated

- Page 3, lines 20-21: These new GNSS navigation satellites, together with planned LEO missions, will offer many more RO observations.

- Page 3, line 22: . . . onboard for the first time. . .

- Page 3, line 29: will have GNOS on board as well, similar . . .

- Page 3, line 30: The definition for the acronym GRAS is defined on page 16, should be where it is first used.

- Page 4, line 1-3: This description is a bit confusing. You mention three antennas on the instrument, then an antenna for the processor that has a stable phase center. Is this one of the three antennas? Or an additional antenna? Please consider rewording.

- Page 4, line 6: Can you quantify "large"? Perhaps by the number of days or occultations

- Page 4, line 16: Should "GPS" be changed to "GNSS"? Single differencing may have been limited to GPS in your references, but here you use GNSS elsewhere in the same sentence.

[Figure]

- Page 4, line 21: Can you reword "started to be used"?

- Page 4, line 24-25: . . .by an ultra-stable oscillator that, so far, was only available for GRACE . . .

- Page 4, line 29: So far, BDS can provide good regional coverage . . .

- Pages 4-5, lines 31, 1-2: . . . GNOS satellite received signals from five geostationary orbit (GEO) satellites, five inclined geosynchronous orbit (IGSO) satellites, and four medium earth orbit (MEO) satellites to conduct the radio occultation measurements.

- And throughout the paper, don't redefine GEO, IGSO, and MEO. Define the first time, and use the acronyms thereafter.

- Page 5, line 7: Remove the word "anyway"

- Page 8, lines 6-7: . . .as the basic equation and adopt Eq. (2) as the auxiliary equation.

- Page 9, line 2: Reword "comparing with" (could use "as compared to")

- Page 9, line 6: . . . constellation, as with the current BDS. In addition, zero-differencing will likely . . .

- Page 9, lines 6-10: Please consider splitting this sentence into multiple sentences.

- Page 9, line 11: In the zero-differencing approach, we employ. . . (the term Zero-Differencing is used previously in the paper. If you want to use it as an acronym, please define earlier).

- Page 9, line 12: "GPS" should be "GNSS", right?

- Page 9, line 21: When you say that the processing chooses the GNSS satellite with highest elevation angle, are you using both GPS and BDS satellites for single-differencing? Please clarify.

- Page 10, line 4: For both B1 and B2, the elevation angle appears to be more like 12 deg where the carrier phase errors are less than 2 mm.

- Page 13, line 10: It looks like you might be missing a reference here.

- Page 13, line 12: MEO is already defined previously in the paper.

- Page 16, lines 4-5: Should be Allan deviation (ADEV), not Allen variance.

---

## Referee Comment (RC2) · Anonymous Referee #2 · 11 Jul 2017

This paper presents results from the "GNOS" radio occultation (RO) measurements aboard the Chinese FY-3C satellite. It is shown that BeiDou GNSS observations, analyzed in single-differencing (SD) and zero-differencing (ZD) mode, produce bending angle and refractivity profiles of equivalent quality, when compared to ECMWF and co-located radiosonde data. In addition, due to the non-uniform global coverage of the current BeiDou space segment the ZD data set includes about 20% more events compared to the SD set because occasionally suitable BeiDou satellites providing the reference link were not available within the receiver's antenna field-of-view. Furthermore, a unique feature of the BDS system is that the signal transmitters are placed into three diverse orbits (MEO, IGSO, GEO). The present study convincingly shows that these orbit differences significantly modify the zonal and meridional distribution of RO events, but have no appreciable impact on the quality of the derived atmospheric profiles.

This well-written paper is a valuable contribution to the present knowledge on single-versus zero-differencing RO analysis and I definitely recommend publication with some minor modifications described below.

General comments:

As emphasized by the authors the successful application of zero-differencing is made possible by the presence of an ultra-stable oscillator driving the GNOS instrument. It would be instructive to illustrate the performance of this clock by providing clock offset statistics. These could be extracted from the results of the FY-3C precise orbit determination.

The comparisons of SD and ZD with ECMWF and radiosonde data are instructive and illuminating. In addition, the direct comparison between SD and ZD bending angle profiles would be worthwhile to consider, in order to substantiate the hypothesis that no biases between the SD and ZD results exist. If possible, I would encourage the authors to add a corresponding figure in the revised paper.

Specific remarks and questions:

Page 3, lines 21ff:

> "One of these LEO missions is China's GNss Occultation Sounder (GNOS) onboard first time on the FengYun 3 series C satellite (FY-3C), [...]."

For completeness I suggest to add the reference

> Bai, W. H., Sun, Y. Q., Du, Q. F., Yang, G. L., Yang, Z. D., Zhang, P., Bi, Y. M., Wang, X. Y., Cheng, C., and Han, Y.: An introduction to the FY3 GNOS instrument and mountain-top tests, Atmos. Meas. Tech., 7, 1817–1823, 10.5194/amt-7-1817-2014, 2014.

Page 4, lines 6–7:

> "So far, a large dataset of FY-3 GNOS RO observations has been obtained."

If I understand correctly, GNOS measurements aboard FY-3C started in September 2013. Thus, as of now the available data set should cover more than 3.5 years. I suggest to add a comment clarifying the decision to restrict the data analysis to the time period of three months between October and December 2013.

Page 6, lines 9ff:

> "Specifically, in this study, we use the BDS satellite data as orbital data inputs and outputs, while time-wise also using GPS time for the processing of the BDS data."

I'm not sure I understand this sentence. Is GPS time used for time-tagging of GPS as well as BDS observations? Please explain.

Page 6, eqn. (1), page 7, eqn. (2), and elsewhere:
To avoid a potential misunderstanding, I suggest to define $\delta t_a$ as the LEO clock error (offset) *at the time of signal reception* and similarly $\delta t^b$ as the GNSS clock error (offset)

[Figure]

*at the time of signal transmission*. With this change there is no need to regard $\delta t_a$ and $\delta t^b$ as functions and the function arguments in brackets (which might be confused with brackets marking an algebraic expression) could be dropped.

Page 7, lines 13ff:

> "The GNSS satellite orbits (positions and velocities) and the GNSS clock offset estimates [...] are provided by the International GNSS Service [...]."

IGS orbits are provided in a terrestrial reference frame. Here, a (quasi-)inertial true-of-date frame (page 5, section 2.1 "Basic algorithm of the excess phase processing") is used. For clarity, I suggest to add a remark indicating that a corresponding frame transformation has been applied.

Page 8, eqn. (7) and (8):
Which one of the two equations is used in the actual processing?

Page 9, lines 19ff:

> "In order to use that specific reference satellite that most likely has the best signal quality and lowest ionospheric influence, our FY-3C GNOS processing chooses the GNSS satellite with highest elevation angle as the reference satellite."

From Bai et al. (2014) (see reference above) I had assumed that the decision which satellite to track as reference is already taken at the receiver level and not during data processing. Second, it would be interesting to note if the reference satellite is tracked by the occultation or zenith antenna. In the latter case SNR at high elevation angles is

expected to be higher at the expense of an additional attitude dependence which must be corrected for. Please clarify.

Page 9, lines 19ff:

> "In practice, less than 0 deg means that there is in fact no reference satellite in view and [...]"

At a (sun-synchronous) orbit height of about 840 km (reference) satellites at elevation angles down to $-27°$ could indeed be visible. Please clarify and/or rephrase the sentence.

Page 10, line 24:

> "In our data processing, a quality control algorithm has been used."

I suggest to quote the fraction of RO events removed by quality control.

Page 12, lines 15ff:

> "The target domain for the comparative statistical analysis is from 5 km to 35 km height [...], since commonly the data quality above 35 km and below 5 km is less good, due to the ionospheric effects and tropospheric multipath effects, respectively [...]"

I assume that the data retrieval is based on geometric optics and wave optical methods (CT, FSI) have not applied. Please clarify.
Pages 25 & 26, Figs. 6 & 7:
From the figure inserts it appears that the analysis is based on the intersection of the SD and ZD data sets and that the intersection contains less events than both, the SD and ZD data set. Why are there 192 (if I counted correctly) events found in the (quality-controlled) SD data set, which did not make it into the ZD set? I suggest to add a clarifying remark.

Page 26 & 27, Figs. 7 & 8:
Why is geopotential height instead of geometric height used as vertical coordinate? Please clarify.

Technical corrections:

Page 7, eqn. (2):
Typo: $\delta\rho^{b}_{a,ion,i}$ $\rightarrow$ $\delta\rho^{c}_{a,ion,i}$

Page 7, eqn. (3):
Typos: $\rho^{a,c}_{a}$ $\rightarrow$ $\rho^{b,c}_{a}$
and the three bracketed expressions need to be squared.

Page 7, eqn. (4):
I suggest to replace the horizontal bars in eqn. (4) ($\overline{r^{b,c}}$ and $\overline{v^{b,c}}$) by a more conventional notation indicating vectors, e.g., $\vec{r}^{\,b,c}$ and $\vec{v}^{\,b,c}$ or $\mathbf{r}^{\,b,c}$ and $\mathbf{v}^{\,b,c}$ (bold-faced font).

Page 8, eqn. (6):
Here, in contrast to eqn. (4), the horizontal bar seems to differentiate between

transmitter and receiver dipole vector. I suggest to clarify the notation.

Page 13, line 10:
There appears to be a reference missing (empty bracket).

---

## Referee Comment (RC3) · Anonymous Referee #3 · 3 Aug 2017

This paper introduces, in a comprehensive way, the data processing of the first Beidou-based Chinese radio occultation mission - FY-3C GNOS and 3-month data were used for the study/data processing. The two strategies of data processing investigated are zero-differencing and signle differencing. Differencing is a standard data process strategy in GNSS data process to mitigate (or cancel out) the various errors (e.g. signal generation/emmision, signal propogation, signal transmission and signal reception) inherited with the technology. Vairous analyses of the atmospheric profiles based on the single- and zero-differencing data processing strategies and using three months'

data, are carried out to evaluate the quality of BDS GNOS RO data and the robustness/quality of the zero-differencing data processing method. By comparing with ECMWF model and co-localated radiosonde data, the BDS GNOS atmospheric profiles derived are fairly consistent.

Data processing algorithms are introduced in a fairly detailed way. The analyses are described and presented in a logical and clear manner. The discussions are comprehensive albeit some further clarification is needed. The conclusions given from the analysis are sound and reflect the current state-of-the-art in the field.

Following are my other comments/suggestions for correction

1) FY-3 C GNOS receivers can receive both the GPS and BDS signals for navigation and occultation modules, therefore GNOS provides a different way to validate its BDS RO data (i.e. based on the zero-difference processing and GPS RO retrievals). I wonder the reason why not to use the GPS GNOS RO retrievals to validate BDS's counterparts?

2) The current coveragte of Beidou is regional. It would be great if the authors can comment over the issue of limited coverage of the Beidou system and how it affects the ROE occurence?

3) Technical Corrections - Define the acronym for GRAS, GEO, IGSO and etc. when they appear in the first place in the text and use the acronyms thereafter. - Page 3, lines 13-14: the word "satellites" is repeated - Page 13, line 10: It looks like you might be missing a reference here. - Page 16, lines 4-5: Should be Allan deviation (ADEV), not Allen variance. - Be careful with some reference formats and typos. - be careful in using the differential technique, you need to be consistent to use differencing or differenced or difference. They do have minor differences. The "single-different" in figure 5 (a)/(b) is NOT right. - the title of the paper looks awkward and it needs to change "processing" and "data" need to be "data processing" - GNSS is commonly referred to The Global Navigation Satellite Systems (plural!!!) - the language usage

neess to be sharpened and grammatical problems are spotted. - "sub-global" needs to be replaced as "regional"

---

## Referee Comment (RC4) · Anonymous Referee #4 · 11 Aug 2017

General comments:

The paper describes the evaluation of zero-difference processing vs. single-difference processing of BeiDou System (BDS) radio occultation (RO) data collected by the Chineese GNOS instrument on board the FY-C3 satellite.

Although this is not the first paper describing GNOS BDS retrievals, it is the first paper describing the zero- and single-differencing methods in that context, and comparing results using either method. For that reason it should be published in AMT. There are,

however, a number of issues that needs to be addressed in a revised version.

The single- and zero-differencing methods are outlined and their application seems sound, although not all relativistic corrections seem to be adequately described. This, together with small unexpected differences in the results, gives me a grain of uncertainty as to whether the relativistic effects and clock offsets are correctly removed. I elaborate on this in one of the specific comments below.

Comparisons of derived bending angle and refractivity to reference profiles from ECMWF analyses are encouraging, although I do not think the results presented show that GNOS BDS RO data are of such high quality as claimed in the text. The authors mention (top of page 11) that part of the bias in their results could be from differences in vertical geolocation of GNOS and reference profiles (which are from ECMWF analyses and radiosondes). I'm not sure that such differences could give rise to the biases that are shown, but if so, such systematic difference should be better understood in the presented data set and possibly corrected. I am not aware of bias problems in the data evaluation of GPS RO data from other sources, e.g., COSMIC/CDAAC or Metop/EUMETSAT. In my comments to the results in Fig. 6 and 7 below, I point to a few other issues that are not mentioned in the text, but which should at least be discussed if improvement of the results in a revised manuscript is not possible.

If improvement is not possible, then some of the statements in the paper should be toned down, e.g., in the abstract where it says that "The statistical evaluation against these reference data shows that the results from single- and zero-difference processing are consistent in both bias and standard deviation, clearly demonstrating the feasibility of zero-differencing for GNOS BDS RO observations.", or at the end of the abstract where it says "The validation results establish that GNOS can provide, on top of GPS RO profiles, accurate and precise BDS RO profiles both from single- and zero-difference processing." Although the GNOS BDS RO data might be of a very high quality comparable to that of GPS RO data, such claims are not fully supported by the results in this paper.

One thing that could give more confidence in the single- and zero-differencing results would be to show cases of ionospheric corrected excess phases at very high altitudes. Although the ionospheric correction in the processing is done at the bending angle level (at common impact parameters of B1 and B2 bending angles), excess phase data corrected at the same times could be shown for cases where the ionospheric residual is small (this could be based on the difference between B1 and B2 excess phases, choosing only cases where such difference/variation is small). If such cases, at altitudes above ∼100 km, show virtually no slope (giving confidence that the relativistic effects and clock offsets are correctly removed), and only noise at the level indicated in Fig. 3, then that would give added confidence in the quality of the data. A few examples together with statistical evidence that ionospheric corrected excess phases at high altitudes are virtually flat compared to the random noise, would make a very good case.

Unfortunately, there is no method description of the derivation of bending angle and refractivity. Could such description be added (possibly just with reference to previous works)?

A few additional questions comes to mind: Are there both setting and rising occultations in the statistics, and how many of each? How far down is the B2 signal typically tracked in rising and setting? How far up are the signals typically tracked? Are extrapolation of B1-B2 performed in the troposphere to extend profiles down to where B1 is tracked (if it is tracked lower than B2)?

Below I give specific comments and technical corrections with <page>/<line> referring to the pdf copy of the manuscript. In some places I give suggestions for improved language that could easy the readability, but not in all places where such improvement could be warranted. Suggested words are in square brackets. I kindly urge the authors to run the manuscript by a person with excellent skills in the English language.

Specific comments and Technical corrections:

1/3: Consider a small change to the title: "... data [from] the FY-3C GNOS mission"

1/22: "[The] GNOS ..."

1/26: "... on [the] FY-3C GNOS, [and] thus ..."

2/12: Skip "as small as".

2/13-14: Bad syntax: "including for the GEO, IGSO, and MEO subsets.". Could be skipped here, since you already indicated earlier in the abstract that the data are from these three sub-systems.

2/14-15: "as may be expected from its lower vulnerability to noise." could also be skipped here.

2/17: "... satellites [can] thus provide..."

2/24-26: Move "Earth's" to before "atmospheric parameters...".

3/9: "LEO" is not previously defined.

3/20-23: I suggest reformulation, e.g.: "One of these LEO missions is the FengYun 3 series C satellite (FY-3C), carrying China's first GNSS Occultation Sounder (GNOS) (Liao et al., 2016). FY-3C was successfully launched on 23 September 2013."

3/28: I suggest reformulation, e.g.: "... satellites, the next being FY-3D, scheduled for launch in 2017, will also carry GNOS instruments, similar to ..."

4/1-3: Please reformulate. Are the antennas considered part of the instrument (line 1) or are they used by the instrument (lines 2-3)?

4/4: "... in [the] GNOS design.".

4/5: "... from Earth's surface ..."

4/13: Replace "it" with "the single-difference method".

4/15: "... [the] single-difference ..."

[Figure]

4/15-16: Redundant information (and bad syntax) that could be skipped: "during the GPS clock offset estimation process."

4/17: "... needs no ground station data, [the] processing is simpler".

4/24: "...requires that the LEO receiver [is equipped with] an ...".

4/26: "... is [equipped with] such ...".

4/31: "... received [the signals from five] geostationary ... (MEO) orbit satellites.".

Section 1: Perhaps you could mention the B1 and B2 frequencies somewhere in the introduction.

Section 1: Perhaps you could mention the different semimajor axes and inclination of the GEO, IGSO, and MEO sub-systems somewhere in the introduction.

5/23: "Recently, [because of] its higher complexity ...".

6/1: "The inputs to [the processing] ...".

6/20: "... for [the] receiver clock and [the] GNSS satellite clock ...".

6/9-11: I do not understand this sentence: "Specifically, in this study, we use the BDS satellite data as orbital data inputs and outputs, while time-wise also using GPS time for the processing of the BDS data." Please clarify.

6/13: "(in units of [length]) at [carrier signal] i". (also in first line of page 7)

6/16: Is there a reference for eq. (1)?

6/17: Skip "(m/s)".

7/4: Superscript on last term in eq. (2) should be "c".

7/10: Superscript "a" should be "b" on the left-hand side of eq. (3).

7/12: Shouldn't it be capital letters "B or C" here?

7/17: Please provide a reference for eq. (4). You say that this is a "periodic relativistic effect", but does not mention the main part of the relativistic correction, and it is therefore unclear if you make all the necessary corrections. If I understand relativistic effects in the GPS correctly, then eq. (4) is a residual that comes about because the GPS transmitters have their clocks adjusted prior to launch, such that the GPS clocks in orbit beat at the same rate as a clock on the Earth (Ashby, Relativity in the Global Positioning System, Living Rev. Relativity, 6, (2003), 1, http://www.livingreviews.org/lrr-2003-1). However, part of that adjustment of the transmitter clock results in an additional frequency shift in ECI that must be taken into account in the zero-differencing (it cancel in the single-differencing). The shift is proportional to the effective gravitational potential at the surface of the rotating Earth, and is actually larger then the relativistic effect in orbit that would have been without this clock adjustment. See, e.g., eq. (46) in Ashby (2003). You should make clear how you make this additional correction. Ashby describes the relativistic effects in the GPS. Are clocks in the BDS similarly adjusted before launch? If so, it would be interesting if you could give the different values of the frequency adjustments in the BDS subsets (GEO, IGSO, and MEO). Also, eq. (4) is relevant for GNSS clocks (as you write), but what about the relativistic effects of the FY-3C satellite clock? They do not seem to be described? Nor is it mentioned how they are estimated in the zero-differencing. Again, eq. (46) in Ashby (2003) could be of help here. In any case, you should make clear how you estimate all the main relativistic effects and clock offsets (please also make clear whether you consider the correction for the transmitter clock adjustment part of the clock offset or part of the relativistic effects).

7/18: Shouldn't there be bars above r and v here?

7/18: Shouldn't it be "GNSS" instead of "GPS"?

7/20: Please provide a reference for eq. (5).

7/20: Subscript "r" should be "a" five places in eq. (5).

8/2: I think eq. (6) needs to be multiplied by the i'th wavelength to be consistent with the terms in eq. (1) and (2).

8/3: In eq. (4) bars were used to indicate vectors, so it is unfortunate here to distinguish the two effective dipole vectors by a bar on one, but not the other. I suggest to use another distinction and consistently use bars to indicate vectors.

Section 2.1: Generally, It would make good sense to mention the order of magnitude of different corrections and their relative importance.

8/6: I suggest skipping "adopt".

8/10: I suggest to remove "employing Eq. (2),".

8/14-16: Some of the "a" and "c" subscripts and superscripts on the right-hand sides of eq. (7) and (8) should be interchanged.

8/14-16: I suggest the use of different symbols in eq. (7) and (8) (and similar in eq. 9) for the phases on the right-hand side, since these are corrected for the effects mentioned in the first paragraph of this section. Perhaps you could simply use a tilde to indicate that they are not strictly the same as the ones in eq. (1) and (2), and at the end of the first paragraph in section 2.2 (line 9) you could write something like: "In the following we refer to these as <symbol_ab> and <symbol_ac>, respectively."

8/18: I suggest replacing ". c1 and c2 are just" with "are".

9/18: You could say "mentioned" instead of "aforementioned".

9/21: Could the occulting and reference satellites be from two different sub-systems, e.g., a MEO as reference for an occulting GEO?

10/1-2: It is not clear how you calculate the "carrier phase observation error standard deviation" shown in Fig. 3. Are you applying a high-pass filter? With what band-width?

11/24 (and other places): You could use the word "difference" instead of "error".

[Figure]

11/25: Use a mathematical symbol for bending angle in eq. (11) (BA is an abbreviation, not a symbol).

11/28: You could here introduce the use of "Bias" and "StdDev" as they are used later in the text: "... estimates of biases (Bias) and standard deviations (StdDev) are illustrated ...".

12/27-28: I do not understand the sentence in parenthesis: "though more standard deviation suppression might be expected from avoiding the reference link computation". It is not clear what "standard deviation suppression" mean, and I'm not sure if this statement is different from what you just said in the sentence before? Using the word "though" indicates that it is contradicting what you said before. Please clarify.

12/28: Schreiner et al. 2009 is not in the reference list. Should perhaps be 2010.

13/3: "Scherllin" instead of "Scherrlin".

13/10: Empty parenthesis. Perhaps a reference is missing.

13/11-12: Is it really the first time that RO retrievals from other than the BDS MEO is demonstrated? Liao et al. (2016) also describes the GNOS-BDS occultation coverage using BDS GEO and IGSO, and I could not find any indication in their paper that the statistics they show is only from MEO occultations. If it is the first time, then you should here make clear that the results in Liao et al. (2016) did not include GEO and IGSO occultations.

14/3: You say that GNOS BDS retrievals are comparable to GPS retrievals, but you have not really shown that here. Either you should show comparisons to GPS retrievals, or you need to support such statements with citations to previous works.

14/5: "... not [only] on MEO satellites but [also] on GEO and IGSO satellites.".

14/26: You say that "Single-differencing does not need to correct the receiver clock offset". I know what you mean, but it is not strictly correct. The receiver clock offset is

removed because it cancel in the single-differencing. Please reformulate.

15/2-4: You say that in the zero-differencing there can be some residual errors after the clock offset correction, but you have not shown that anywhere. Can you give examples of such residuals?

16/4: It should be "Allan", not "Allen".

16/5: Please reformulate the statement on the Allan variance (or deviation) here. It is correctly formulated in the abstract. The unit is not second.

16/10-12: The last paragraph should be reformulated or removed. It is unclear what "in this context of the leading instruments" means.

Results shown in Figure 6:

1) StdDev: I would have expected visible differences between single- and zero-differencing to be only at high altitudes/impact heights. However, even below 10 km it seems that the StdDevs are significantly different, with the zero-differencing results generally having the larger StdDev. How can that be explained? From the legend it appears that it is exactly the same number of occultations involved (and I assume therefore that it is the same occultations). Also, it seems that the StdDev starts increasing already at 20-25 km. I would have expected the increase to start a bit higher when I compare with GPS RO statistics from other sources.

2) Bias: The differences in bias between single- and zero-differencing are similar below 20 km, and the somewhat larger negative bias for GEO can probably be explained by the fact that all the GEO occultations are at very high altitudes (and for some reason that gives a larger negative bias when compared to ECMWF). However, above 20 km, the biases for the three subsystems (MEO, IGSO, GEO) are diverging more for the zero-differencing results, and in particular the bias for the GEO occultations becomes more negative than it is for single-differencing. Why?

These issues needs to be discussed in the text.

Results shown in Figure 7:

The same comments as above applies here, but additionally, it is very strange to see the bias for the GEO occultations for single-diffencing at high altitudes being more positive than the others. This is inconsistent with the biases in the bending angle. It is critically important to understand this, since you are trying to make the point that zero-differencing has lower StdDev than single-differencing, but it is difficult to have confidence in the results if there are such inconsistencies in the biases.

Figure 8 axes labels: I suggest to redo this figure with labels as in Figure 7 ("R%" does not make sense; "geop" should be "Geopotential height").

---

## Author Comment (AC1) · 13 Nov 2017

We thank the referees very much for the constructive comments and recommendations and for the overall positive rating that this is a significant scientific paper. We thoroughly considered all comments and carefully revised the manuscript accounting for most of them. In addition, we carefully complemented these revisions with a range of further improvements throughout the manuscript text in the spirit of the comments.

(Please read the amt-2017-177-supplement.pdf by the link at end of this document, in

which you can find the response to all the referees and the revised manuscript)

1 General Comments

This article presents atmospheric profiling results derived from radio occultation of satellites in the BeiDou constellation over three months, as processed by single- and zero- differencing algorithms. The derived bending angles and refractivity profiles are compared to results from the ECMWF and radiosondes collocated geospatially and temporally with the profiles from the GNOS instrument. This paper is well organized and does a good job of describing the processing methodologies. The resulting BDS profiles are fairly consistent with both other radio occultation measurement results from the ECMWF and localized radiosondes. The results are encouraging in both the use of ultra-stable oscillators for radio occultation collection instruments (zero-differencing), and the use of the BeiDou signals as remote sensing sources for future atmospheric sensing satellite missions. I only have a few specific comments and suggestions that I'd like to see further expanded upon in the revision.

Thank you.

2 Specific Comments

The authors mention, early in the paper, that the GNOS receiver is capable of collecting both GPS and BDS data. However, I am slightly confused as to whether any GPS data were used in your single/zero-differencing studies. You make a distinction on Page 6 that the term "GNSS" refers to both GPS and BDS satellites, but it seems like only BDS satellites are used for the occultation measurements, and perhaps GPS is just used for timing? It would be interesting to the reader to compare occultation results from your same algorithms, but with GPS data over the same time and spatial intervals.

Ok, though it is a challenge to get sufficient co-located BDS and GPS radio occultation (RO) profiles in our current setup, we now performed some comparative RO data processing of BDS vs. GPS satellite observations by using the single-/zero-differencing

algorithms as well. We note that evaluations of the retrieved GPS-only RO data using single-differencing algorithms have been presented by some previous papers already, so that is why this paper focuses on the BDS RO data validation. We included one BDS vs. GPS intercomparison figure now in section 3.2., which we find to exhibit reasonably high consistency. Of course, further improvements and a detailed intercomparison analysis of the GPS and BDS RO data is a very interesting study for us as well, and we plan to do it by an extra paper. And yes, the GPS is used for timing in the BDS data processing, as described in section 2.1.

Another related point, I am curious as to why your results are negatively biased from both the ECMWF and radiosondes. The authors make a comment as to the differences in the vertical geo-locations of the profiles in comparison to the reference data, but it is odd that all the different types of BDS satellites (GEO, IGSO, MEO) are negatively biased. Again, if the authors were to process GPS data from the same times/locations with their single/zero-differencing algorithms, it could be another way to validate their methodologies and results. The negative biases are already quite small but, yes, we agree we should be able to further reduce them in future. Currently we consider they are likely caused by a residual error in the excess phase processing and we work to further improve this processing.

The authors use radiosonde measurements within a +/- 1 deg lat-lon/ +/- 1 hour collocation criterion to validate an RO event for part of their analysis. This range can be on the order of a 200 km x 200 km box, over the course of an hour. Do the authors have an explanation or reference to the stability of the atmosphere over these spatial and time ranges?

Thanks, its a good question. Actually, the $\pm 1$ degree lat-lon and $\pm 1$ hour criterion was our initial collocation implementation, and we used it at the beginning of GNOS data validation. We have re-checked our programming codes, and confirmed that the temporal and spatial criterion of comparison between the GNOS BDS RO observations and the radiosonde reference data is within $\pm 1$ hour and a circle with radius of 200

km around the radiosonde location. So we have corrected the explanation of the collocations accordingly. Regarding the reasonable stability of the atmosphere over such collocation distances, we have now included Hajj et al. (2004) and Anthes et al. (2008) as references, since therein some discussion of representativity errors as a function of collocation distances is conducted.

3 Technical Corrections

Page 3, lines 13-14: the word "satellites" is repeated Ok, corrected.

Page 3, lines 20-21: These new GNSS navigation satellites, together with planned LEO missions, will offer many more RO observations. Ok, done.

Page 3, line 22: : : : onboard for the first time: : : Ok, done.

Page 3, line 29: will have GNOS on board as well, similar : : : Ok, done.

Page 3, line 30: The definition for the acronym GRAS is defined on page 16, should be where it is first used. Ok, done.

Page 4, line 1-3: This description is a bit confusing. You mention three antennas on the instrument, then an antenna for the processor that has a stable phase center. Is this one of the three antennas? Or an additional antenna? Please consider rewording. It is one of the three antennas, but not an additional antenna. The 'as well as' has been revised as 'in which'.

Page 4, line 6: Can you quantify "large"? Perhaps by the number of days or occultations Ok, done. The 'large' has been revised as '4-year'.

Page 4, line 16: Should " GPS" be changed to "GNSS"? Single differencing may have been limited to GPS in your references, but here you use GNSS elsewhere in the same sentence. Thanks, it should be 'GPS', since this specifically refers to the GPS 'selective availability' (SA).

Page 4, line 21: Can you reword "started to be used"? Ok, we now say "was started to

be used"

Page 4, line 24-25: : : :by an ultra-stable oscillator that, so far, was only available for GRACE : : : Ok, done.

Page 4, line 29: So far, BDS can provide good regional coverage : : : Ok, done.

Pages 4-5, lines 31, 1-2: : : : GNOS satellite received signals from five geostationary orbit (GEO) satellites, five inclined geosynchronous orbit (IGSO) satellites, and four medium earth orbit (MEO) satellites to conduct the radio occultation measurements. Ok, done.

And throughout the paper, don't redefine GEO, IGSO, and MEO. Define the first time, and use the acronyms thereafter. Ok, done.

Page 5, line 7: Remove the word "anyway" Ok, done.

Page 8, lines 6-7: : : :as the basic equation and adopt Eq. (2) as the auxiliary equation. Ok, done.

Page 9, line 2: Reword "comparing with" (could use "as compared to") Ok, done.

Page 9, line 6: : : : constellation, as with the current BDS. In addition, zero differencing will likely : : : Ok, done.

Page 9, lines 6-10: Please consider splitting this sentence into multiple sentences. Ok, done; split into two sentences.

Page 9, line 11: In the zero-differencing approach, we employ: : : (the term Zero-Differencing is used previously in the paper. If you want to use it as an acronym, please define earlier). Ok, done. Page 9, line 12: "GPS" should be "GNSS", right? Ok, corrected.

Page 9, line 21: When you say that the processing chooses the GNSS satellite with highest elevation angle, are you using both GPS and BDS satellites for singledifferencing? Please clarify. Currently, in our single-differencing data processing, only BDS satellites are used as reference satellites for BDS occultation, similarly only GPS satellites for GPS occultation. We clarified this in the text now. Page 10, line 4: For both B1 and B2, the elevation angle appears to be more like 12 deg where the carrier phase errors are less than 2 mm. Right, as shown in Figure 3, for both B1 and B2, the elevation should be 12 deg, where the carrier phase errors are less than 2 mm. As well as, at 10 degree both the B1 and B2 carrier phase errors are less than 2.2 mm. Actually, we use the elevation 10 degree as the reference satellite selection criterion, so we have revised the 2 mm to 2.2 mm in the manuscript.

Page 13, line 10: It looks like you might be missing a reference here. Thanks, was left as a typo, corrected.

Page 13, line 12: MEO is already defined previously in the paper. Ok, corrected.

Page 16, lines 4-5: Should be Allan deviation (ADEV), not Allen variance. Ok, corrected.

Please also note the supplement to this comment:
https://www.atmos-meas-tech-discuss.net/amt-2017-177/amt-2017-177-AC1-supplement.pdf

---

## Author Comment (AC2) · 13 Nov 2017

We thank the referees very much for the constructive comments and recommendations and for the overall positive rating that this is a significant scientific paper. We thoroughly considered all comments and carefully revised the manuscript accounting for most of them. In addition, we carefully complemented these revisions with a range of further improvements throughout the manuscript text in the spirit of the comments.

(Please read the amt-2017-177-supplement.pdf by the link at end of this document, in

which you can find the response to all the referees and the revised manuscript)

This paper presents results from the "GNOS" radio occultation (RO) measurements aboard the Chinese FY-3C satellite. It is shown that BeiDou GNSS observations, analyzed in single-differencing (SD) and zero-differencing (ZD) mode, produce bending angle and refractivity profiles of equivalent quality, when compared to ECMWF and co-located radiosonde data. In addition, due to the non-uniform global coverage of the current BeiDou space segment the ZD data set includes about 20% more events compared to the SD set because occasionally suitable BeiDou satellites providing the reference link were not available within the receiver's antenna field of view. Furthermore, a unique feature of the BDS system is that the signal transmitters are placed into three diverse orbits (MEO, IGSO, GEO). The present study convincingly shows that these orbit differences significantly modify the zonal and meridional distribution of RO events, but have no appreciable impact on the quality of the derived atmospheric profiles. This well-written paper is a valuable contribution to the present knowledge on single versus zero-differencing RO analysis and I definitely recommend publication with some minor modifications described below.

Thank you.

General comments: As emphasized by the authors the successful application of zero-differencing is made possible by the presence of an ultra-stable oscillator driving the GNOS instrument. It would be instructive to illustrate the performance of this clock by providing clock offset statistics. These could be extracted from the results of the FY-3C precise orbit determination.

Ok, it's a good advice, a detailed comparison analysis of the ZD and SD algorithms is a very interesting study point for us as well, and we plan to do it by an extra paper. For this paper, we preferred to give a concise algorithms description and focus on our initial FY-3C GNOS data evaluation and validation.

The comparisons of SD and ZD with ECMWF and radiosonde data are instructive and

illuminating. In addition, the direct comparison between SD and ZD bending angle profiles would be worthwhile to consider, in order to substantiate the hypothesis that no biases between the SD and ZD results exist. If possible, I would encourage the authors to add a corresponding figure in the revised paper.

Ok, it's a good idea to show the direct comparison between SD and ZD bending angle and other retrieved profiles to substantiate the consistency of the SD and ZD results (but not the hypothesis of strictly no biases between the SD and ZD results, we believe, because if the LEO satellite clock is stable and accurate enough, the ZD results should be with higher accuracy than the ZD results, theoretically). On the other hand, the topic of this paper is 'evaluation of atmospheric profiles derived from single- and zero-difference excess phase processing of BeiDou System radio occultation data of the FY-3C GNOS mission', but not comparison analysis of ZD and SD algorithms. Therefore, we preferred to keep the comparisons of SD and ZD with ECMWF and radiosonde data so far, since those figures are very helpful to provide an initial evaluation and validation of the SD and ZD retrievals in a scientifically reasonable way. Moreover, the readers somehow can see the level of consistency of the SD and ZD retrievals through these comparison figures. Considering the main topic and space limitation of this paper, we therefore preferred to keep the current comparison strategy and figures (and leave rigorous SD, ZD intercomparisons as next steps of refined analyses).

Specific remarks and questions: Page 3, lines 21ff: "One of these LEO missions is China's GNss Occultation Sounder (GNOS) onboard first time on the FengYun 3 series C satellite (FY-3C), [...]." For completeness I suggest to add the reference Bai, W. H., Sun, Y. Q., Du, Q. F., Yang, G. L., Yang, Z. D., Zhang, P., Bi, Y. M., Wang, X. Y., Cheng, C., and Han, Y.: An introduction to the FY3 GNOS instrument and mountain-top tests, Atmos. Meas. Tech., 7, 1817–1823, 10.5194/amt-7-1817-2014, 2014. Ok, done.

Page 4, lines 6–7: "So far, a large dataset of FY-3 GNOS RO observations has been obtained." If I understand correctly, GNOS measurements aboard FY-3C started in September 2013. Thus, as of now the available data set should cover more than 3.5

years. I suggest to add a comment clarifying the decision to restrict the data analysis to the time period of three months between October and December 2013. Thank you for this suggestion. Right, the available GNOS RO data set is more 3.5 years now. We used the first three month GNOS BDS RO data set in this paper because this period is the GNOS in-orbit testing time, and we have done lots of evaluation and analysis using this dataset. And in our opinion a 3-month GNOS BDS RO dataset is sufficient for in-orbit testing and this initial BDS RO validation paper. Future more climate-oriented analyses will use longer data records.

Page 6, lines 9ff: "Specifically, in this study, we use the BDS satellite data as orbital data inputs and outputs, while time-wise also using GPS time for the processing of the BDS data." I'm not sure I understand this sentence. Is GPS time used for time-tagging of GPS as well as BDS observations? Please explain. Yes, the GPS time is used for time-tagging of GPS as well as BDS observations, as described in Section 2.1.

Page 6, eqn. (1), page 7, eqn. (2), and elsewhere: To avoid a potential misunderstanding, I suggest to define _ta as the LEO clock error (offset) at the time of signal reception and similarly _tb as the GNSS clock error (offset) at the time of signal transmission. With this change there is no need to regard _ta and _tb as functions and the function arguments in brackets (which might be confused with brackets marking an algebraic expression) could be dropped. Ok, done. We have revised the related equations following this criterion, for the clock terms with only a subscript 'a' and only a superscript 'b' or 'c', since we agree this anyway clearly indicates reception time and transmission time. And for the terms with both the subscript 'a' and superscript 'b' or 'c', we just kept the simple argument '(t_r)' to make sure we indicate the allocation to reception time.

Page 7, lines 13ff: "The GNSS satellite orbits (positions and velocities) and the GNSS clock offset estimates [...] are provided by the International GNSS Service [...]." IGS orbits are provided in a terrestrial reference frame. Here, a (quasi-)inertial trueof-date frame (page 5, section 2.1 "Basic algorithm of the excess phase processing") is used. For clarity, I suggest to add a remark indicating that a corresponding frame transformation has been applied. Yes, in our processing, the GNSS satellites' position and velocity information came from IGS orbit products, and then transferred all the position and velocity from ITRF to TOD (ECI) coordination system. We have added such a remark.

Page 8, eqn. (7) and (8): Which one of the two equations is used in the actual processing? Equation (7); we have added this in the text now.

Page 9, lines 19ff: "In order to use that specific reference satellite that most likely has the best signal quality and lowest ionospheric influence, our FY-3C GNOS processing chooses the GNSS satellite with highest elevation angle as the reference satellite." From Bai et al. (2014) (see reference above) I had assumed that the decision which satellite to track as reference is already taken at the receiver level and not during data processing. Second, it would be interesting to note if the reference satellite is tracked by the occultation or zenith antenna. In the latter case SNR at high elevation angles is expected to be higher at the expense of an additional attitude dependence which must be corrected for. Please clarify. Yes, for the FY-3C GNOS, the reference satellite is determined by the software onboard the satellite, and it selects the GNSS satellite with the biggest elevation as a reference satellite. The reference satellite's signal is received by the positioning antenna. We have clarified this in the text now.

Page 9, lines 19ff: "In practice, less than 0 deg means that there is in fact no reference satellite in view and [...]" At a (sun-synchronous) orbit height of about 840 km (reference) satellites at elevation angles down to 27_ could indeed be visible. Please clarify and/or rephrase the sentence. Ok, done.

Page 10, line 24: "In our data processing, a quality control algorithm has been used." I suggest to quote the fraction of RO events removed by quality control. Ok, done.

Page 12, lines 15ff: "The target domain for the comparative statistical analysis is from 5 km to 35 km height [...], since commonly the data quality above 35 km and below 5 km is less good, due to the ionospheric effects and tropospheric multipath effects,

respectively [...]" I assume that the data retrieval is based on geometric optics and wave optical methods (CT, FSI) have not applied. Please clarify. Our RO data processing system from excess phase onwards is based on the ROPP software. So similar to ROPP, our data retrieval is mainly based on the geometric optics (CT), while below 20 km height, both the geometric optics (CT) and wave optical method were used.

Pages 25 & 26, Figs. 6 & 7: From the figure inserts it appears that the analysis is based on the intersection of the SD and ZD data sets and that the intersection contains less events than both, the SD and ZD data set. Why are there 192 (if I counted correctly) events found in the (quality-controlled) SD data set, which did not make it into the ZD set? I suggest to add a clarifying remark. Ok, clarifying remark added in the fig. caption.

Page 26 & 27, Figs. 7 & 8: Why is geopotential height instead of geometric height used as vertical coordinate? Please clarify. We used the geopotential height for Figures 7 and 8, because the data obtained from the ECMWF model and the radiosonde observations used the geopotential height as the vertical coordinate.

Technical corrections:

Page 7, eqn. (2): Ok, done.

Page 7, eqn. (3): and the three bracketed expressions need to be squared. Ok, done.

Page 7, eqn. (4): I suggest to replace the horizontal bars in eqn. (4) (rb;c and vb;c) by a more conventional notation indicating vectors Ok, done.

Page 8, eqn. (6): Here, in contrast to eqn. (4), the horizontal bar seems to differentiate between transmitter and receiver dipole vector. I suggest to clarify the notation. Ok, done.

Page 13, line 10: There appears to be a reference missing (empty bracket). Ok, was a typo left, corrected.

Please also note the supplement to this comment:
https://www.atmos-meas-tech-discuss.net/amt-2017-177/amt-2017-177-AC2-supplement.pdf

———————————————————

[Figure]

**Supplement:**

Manuscript doi: 10.5194/amt-2017-177, 2017.
Manuscript Title: Evaluation of atmospheric profiles derived from single- and zero-difference excess phase processing of BeiDou System radio occultation data of the FY-3C GNOS mission
Authors: Weihua Bai, Congliang Liu, Xiangguang Meng, Yueqiang Sun, Gottfried Kirchengast, Qifei Du, Xianyi Wang, Guanglin Yang, Mi Liao, Zhongdong Yang, Danyang Zhao, Junming Xia, Yuerong Cai, Lijun Liu, and Dongwei Wang

*We thank the referees very much for the constructive comments and recommendations and for the overall positive rating that this is a significant scientific paper. We thoroughly considered all comments and carefully revised the manuscript accounting for most of them. In addition, we carefully complemented these revisions with a range of further improvements throughout the manuscript text in the spirit of the comments.*

*Please find below our point-by-point response (in form of italicized, blue text) to the* reviewers' comments (in form of upright, black text)*, inserted below each comment.*

**Response to Anonymous Referee #1's comments**

**1 General Comments**

This article presents atmospheric profiling results derived from radio occultation of satellites in the BeiDou constellation over three months, as processed by single- and zero- differencing algorithms. The derived bending angles and refractivity profiles are compared to results from the ECMWF and radiosondes collocated geospatially and temporally with the profiles from the GNOS instrument.

This paper is well organized and does a good job of describing the processing methodologies. The resulting BDS profiles are fairly consistent with both other radio occultation measurement results from the ECMWF and localized radiosondes. The results are encouraging in both the use of ultra-stable oscillators for radio occultation collection instruments (zero-differencing), and the use of the BeiDou signals as remote sensing sources for future atmospheric sensing satellite missions. I only have a few specific comments and suggestions that I'd like to see further expanded upon in the revision.

*Thank you.*

**2 Specific Comments**

The authors mention, early in the paper, that the GNOS receiver is capable of collecting both GPS and BDS data. However, I am slightly confused as to whether any GPS data were used in your single/zero-differencing studies. You make a distinction on Page 6 that the term "GNSS" refers to both GPS and BDS satellites, but it seems like only BDS satellites are used for the occultation measurements, and perhaps GPS is just used for timing? It would be interesting to the reader to compare occultation results from your same algorithms, but with GPS data over the same time and spatial intervals.

*Ok, though it is a challenge to get sufficient co-located BDS and GPS radio occultation (RO) profiles in our current setup, we now performed some comparative RO data processing of BDS vs. GPS satellite observations by using the single-/zero-differencing algorithms as well. We note that evaluations of the retrieved GPS-only RO data using single-differencing algorithms have been presented by some previous papers already, so that is why this paper focuses on the BDS RO data validation. We included one BDS vs. GPS intercomparison figure now in section 3.2., which we find to exhibit reasonably high consistency. Of course, further improvements and a detailed intercomparison analysis of the GPS and BDS RO data is a very interesting study for us as well, and we plan to do it by an extra paper.*
*And yes, the GPS is used for timing in the BDS data processing, as described in section 2.1.*

Another related point, I am curious as to why your results are negatively biased from both the ECMWF and radiosondes. The authors make a comment as to the differences in the vertical geo-locations of the profiles in comparison to the reference data, but it is odd that all the different types of BDS satellites (GEO, IGSO, MEO) are negatively biased. Again, if the authors were to process GPS data from the same times/locations with their single/zero-differencing algorithms, it could be another way to validate their methodologies and results.
*The negative biases are already quite small but, yes, we agree we should be able to further reduce them in future. Currently we consider they are likely caused by a residual error in the excess phase processing and we work to further improve this processing.*

The authors use radiosonde measurements within a +/- 1 deg lat-lon/ +/- 1 hour collocation criterion to validate an RO event for part of their analysis. This range can be on the order of a 200 km x 200 km box, over the course of an hour. Do the authors have an explanation or reference to the stability of the atmosphere over these spatial and time ranges?

*Thanks, its a good question. Actually, the $\pm 1$ degree lat-lon and $\pm 1$ hour criterion was our initial collocation implementation, and we used it at the beginning of GNOS data validation. We have re-checked our programming codes, and confirmed that the temporal and spatial criterion of comparison between the GNOS BDS RO observations and the radiosonde reference data is within $\pm 1$ hour and a circle with radius of 200 km around the radiosonde location. So we have corrected the explanation of the collocations accordingly.*
*Regarding the reasonable stability of the atmosphere over such collocation distances, we have now included Hajj et al. (2004) and Anthes et al. (2008) as references, since therein some discussion of representativity errors as a function of collocation distances is conducted.*

**3 Technical Corrections**
Page 3, lines 13-14: the word "satellites" is repeated
*Ok, corrected.*

Page 3, lines 20-21: These new GNSS navigation satellites, together with planned LEO missions, will offer many more RO observations.
*Ok, done.*

Page 3, line 22: : : : onboard for the first time: : :
*Ok, done.*

Page 3, line 29: will have GNOS on board as well, similar : : :
*Ok, done.*

Page 3, line 30: The definition for the acronym GRAS is defined on page 16,
should be where it is first used.
*Ok, done.*

Page 4, line 1-3: This description is a bit confusing. You mention three antennas on the
instrument, then an antenna for the processor that has a stable phase center. Is this one of the
three antennas? Or an additional antenna? Please consider rewording.
*It is one of the three antennas, but not an additional antenna. The 'as well as' has been
revised as 'in which'.*

Page 4, line 6: Can you quantify "large"? Perhaps by the number of days or occultations
*Ok, done. The 'large' has been revised as '4-year'.*

Page 4, line 16: Should " GPS" be changed to "GNSS"? Single differencing may have been
limited to GPS in your references, but here you use GNSS elsewhere in the same sentence.
*Thanks, it should be 'GPS', since this specifically refers to the GPS 'selective availability'
(SA).*

Page 4, line 21: Can you reword "started to be used"?
*Ok, we now say "was started to be used"*

Page 4, line 24-25: : : :by an ultra-stable oscillator that, so far, was only available
for GRACE : : :
*Ok, done.*

Page 4, line 29: So far, BDS can provide good regional coverage : : :
*Ok, done.*

Pages 4-5, lines 31, 1-2: : : : GNOS satellite received signals from five geostationary orbit
(GEO) satellites, five inclined geosynchronous orbit (IGSO) satellites, and four medium earth
orbit (MEO) satellites to conduct the radio occultation measurements.
*Ok, done.*

And throughout the paper, don't redefine GEO, IGSO, and MEO. Define the first time, and use the acronyms thereafter.

*Ok, done.*

Page 5, line 7: Remove the word "anyway"

*Ok, done.*

Page 8, lines 6-7: : : :as the basic equation and adopt Eq. (2) as the auxiliary equation.

*Ok, done.*

Page 9, line 2: Reword "comparing with" (could use "as compared to")

*Ok, done.*

Page 9, line 6: : : : constellation, as with the current BDS. In addition, zero differencing will likely : : :

*Ok, done.*

Page 9, lines 6-10: Please consider splitting this sentence into multiple sentences.

*Ok, done; split into two sentences.*

Page 9, line 11: In the zero-differencing approach, we employ: : : (the term Zero-Differencing is used previously in the paper. If you want to use it as an acronym, please define earlier).

*Ok, done.*

Page 9, line 12: "GPS" should be "GNSS", right?

*Ok, corrected.*

Page 9, line 21: When you say that the processing chooses the GNSS satellite with highest elevation angle, are you using both GPS and BDS satellites for single-differencing? Please clarify.

*Currently, in our single-differencing data processing, only BDS satellites are used as reference satellites for BDS occultation, similarly only GPS satellites for GPS occultation. We clarified this in the text now.*

Page 10, line 4: For both B1 and B2, the elevation angle appears to be more like 12 deg where the carrier phase errors are less than 2 mm.

*Right, as shown in Figure 3, for both B1 and B2, the elevation should be 12 deg, where the carrier phase errors are less than 2 mm. As well as, at 10 degree both the B1 and B2 carrier phase errors are less than 2.2 mm. Actually, we use the elevation 10 degree as the reference satellite selection criterion, so we have revised the 2 mm to 2.2 mm in the manuscript.*

Page 13, line 10: It looks like you might be missing a reference here.

*Thanks, was left as a typo, corrected.*

Page 13, line 12: MEO is already defined previously in the paper.

*Ok, corrected.*

Page 16, lines 4-5: Should be Allan deviation (ADEV), not Allen variance.

*Ok, corrected.*

**Response to Anonymous Referee #2's comments**

This paper presents results from the "GNOS" radio occultation (RO) measurements aboard the Chinese FY-3C satellite. It is shown that BeiDou GNSS observations, analyzed in single-differencing (SD) and zero-differencing (ZD) mode, produce bending angle and refractivity profiles of equivalent quality, when compared to ECMWF and co-located radiosonde data. In addition, due to the non-uniform global coverage of the current BeiDou space segment the ZD data set includes about 20% more events compared to the SD set because occasionally suitable BeiDou satellites providing the reference link were not available within the receiver's antenna field of view. Furthermore, a unique feature of the BDS system is that the signal transmitters are placed into three diverse orbits (MEO, IGSO, GEO). The present study convincingly shows that these orbit differences significantly modify the zonal and meridional distribution of RO events, but have no appreciable impact on the quality of the derived atmospheric profiles.

This well-written paper is a valuable contribution to the present knowledge on single versus zero-differencing RO analysis and I definitely recommend publication with some minor modifications described below.

*Thank you.*

**General comments:**

As emphasized by the authors the successful application of zero-differencing is made possible by the presence of an ultra-stable oscillator driving the GNOS instrument. It would be instructive to illustrate the performance of this clock by providing clock offset statistics. These could be extracted from the results of the FY-3C precise orbit determination.

*Ok, it's a good advice, a detailed comparison analysis of the ZD and SD algorithms is a very interesting study point for us as well, and we plan to do it by an extra paper. For this paper, we preferred to give a concise algorithms description and focus on our initial FY-3C GNOS data evaluation and validation.*

The comparisons of SD and ZD with ECMWF and radiosonde data are instructive and illuminating. In addition, the direct comparison between SD and ZD bending angle profiles would be worthwhile to consider, in order to substantiate the hypothesis that no biases between the SD and ZD results exist. If possible, I would encourage the authors to add a corresponding figure in the revised paper.

*Ok, it's a good idea to show the direct comparison between SD and ZD bending angle and other retrieved profiles to substantiate the consistency of the SD and ZD results (but not the hypothesis of strictly no biases between the SD and ZD results, we believe, because if the LEO satellite clock is stable and accurate enough, the ZD results should be with higher accuracy than the ZD results, theoretically).*
*On the other hand, the topic of this paper is 'evaluation of atmospheric profiles derived from*

*single- and zero-difference excess phase processing of BeiDou System radio occultation data of the FY-3C GNOS mission', but not comparison analysis of ZD and SD algorithms.*

*Therefore, we preferred to keep the comparisons of SD and ZD with ECMWF and radiosonde data so far, since those figures are very helpful to provide an initial evaluation and validation of the SD and ZD retrievals in a scientifically reasonable way. Moreover, the readers somehow can see the level of consistency of the SD and ZD retrievals through these comparison figures.*

*Considering the main topic and space limitation of this paper, we therefore preferred to keep the current comparison strategy and figures (and leave rigorous SD, ZD intercomparisons as next steps of refined analyses).*

**Specific remarks and questions:**

Page 3, lines 21ff:

"One of these LEO missions is China's GNss Occultation Sounder (GNOS) onboard first time on the FengYun 3 series C satellite (FY-3C), [...]."

For completeness I suggest to add the reference

Bai, W. H., Sun, Y. Q., Du, Q. F., Yang, G. L., Yang, Z. D., Zhang, P., Bi, Y. M., Wang, X. Y., Cheng, C., and Han, Y.: An introduction to the FY3 GNOS instrument and mountain-top tests, Atmos. Meas. Tech., 7, 1817–1823, 10.5194/amt-7-1817-2014, 2014.

*Ok, done.*

Page 4, lines 6–7:

"So far, a large dataset of FY-3 GNOS RO observations has been obtained." If I understand correctly, GNOS measurements aboard FY-3C started in September 2013. Thus, as of now the available data set should cover more than 3.5 years. I suggest to add a comment clarifying the decision to restrict the data analysis to the time period of three months between October and December 2013.

*Thank you for this suggestion. Right, the available GNOS RO data set is more 3.5 years now. We used the first three month GNOS BDS RO data set in this paper because this period is the GNOS in-orbit testing time, and we have done lots of evaluation and analysis using this dataset. And in our opinion a 3-month GNOS BDS RO dataset is sufficient for in-orbit testing and this initial BDS RO validation paper. Future more climate-oriented analyses will use longer data records.*

Page 6, lines 9ff:

"Specifically, in this study, we use the BDS satellite data as orbital data inputs and outputs, while time-wise also using GPS time for the processing of the BDS data." I'm not sure I understand this sentence. Is GPS time used for time-tagging of GPS as well as BDS observations? Please explain.

*Yes, the GPS time is used for time-tagging of GPS as well as BDS observations, as described in Section 2.1.*

Page 6, eqn. (1), page 7, eqn. (2), and elsewhere:

To avoid a potential misunderstanding, I suggest to define _ta as the LEO clock error (offset) at the time of signal reception and similarly _tb as the GNSS clock error (offset) at the time of signal transmission. With this change there is no need to regard _ta and _tb as functions and the function arguments in brackets (which might be confused with brackets marking an algebraic expression) could be dropped.

*Ok, done. We have revised the related equations following this criterion, for the clock terms with only a subscript 'a' and only a superscript 'b' or 'c', since we agree this anyway clearly indicates reception time and transmission time. And for the terms with both the subscript 'a' and superscript 'b' or 'c', we just kept the simple argument '(t_r)' to make sure we indicate the allocation to reception time.*

Page 7, lines 13ff:
"The GNSS satellite orbits (positions and velocities) and the GNSS clock offset estimates [...] are provided by the International GNSS Service [...]." IGS orbits are provided in a terrestrial reference frame. Here, a (quasi-)inertial trueof-date frame (page 5, section 2.1 "Basic algorithm of the excess phase processing") is used. For clarity, I suggest to add a remark indicating that a corresponding frame transformation has been applied.

*Yes, in our processing, the GNSS satellites' position and velocity information came from IGS orbit products, and then transferred all the position and velocity from ITRF to TOD (ECI) coordination system. We have added such a remark.*

Page 8, eqn. (7) and (8):
Which one of the two equations is used in the actual processing?
*Equation (7); we have added this in the text now.*

Page 9, lines 19ff:
"In order to use that specific reference satellite that most likely has the best signal quality and lowest ionospheric influence, our FY-3C GNOS processing chooses the GNSS satellite with highest elevation angle as the reference satellite."
From Bai et al. (2014) (see reference above) I had assumed that the decision which satellite to track as reference is already taken at the receiver level and not during data processing. Second, it would be interesting to note if the reference satellite is tracked by the occultation or zenith antenna. In the latter case SNR at high elevation angles is expected to be higher at the expense of an additional attitude dependence which must be corrected for. Please clarify.

*Yes, for the FY-3C GNOS, the reference satellite is determined by the software onboard the satellite, and it selects the GNSS satellite with the biggest elevation as a reference satellite. The reference satellite's signal is received by the positioning antenna.*
*We have clarified this in the text now.*

Page 9, lines 19ff:
"In practice, less than 0 deg means that there is in fact no reference satellite in view and [...]"
At a (sun-synchronous) orbit height of about 840 km (reference) satellites at elevation angles down to 27_ could indeed be visible. Please clarify and/or rephrase the sentence.

*Ok, done.*

Page 10, line 24:
"In our data processing, a quality control algorithm has been used." I suggest to quote the fraction of RO events removed by quality control.
*Ok, done.*

Page 12, lines 15ff:
"The target domain for the comparative statistical analysis is from 5 km to 35 km height [...], since commonly the data quality above 35 km and below 5 km is less good, due to the ionospheric effects and tropospheric multipath effects, respectively [...]" I assume that the data retrieval is based on geometric optics and wave optical methods (CT, FSI) have not applied. Please clarify.
*Our RO data processing system from excess phase onwards is based on the ROPP software. So similar to ROPP, our data retrieval is mainly based on the geometric optics (CT), while below 20 km height, both the geometric optics (CT) and wave optical method were used.*

Pages 25 & 26, Figs. 6 & 7:
From the figure inserts it appears that the analysis is based on the intersection of the SD and ZD data sets and that the intersection contains less events than both, the SD and ZD data set. Why are there 192 (if I counted correctly) events found in the (quality-controlled) SD data set, which did not make it into the ZD set? I suggest to add a clarifying remark.
*Ok, clarifying remark added in the fig. caption.*

Page 26 & 27, Figs. 7 & 8:
Why is geopotential height instead of geometric height used as vertical coordinate? Please clarify.
*We used the geopotential height for Figures 7 and 8, because the data obtained from the ECMWF model and the radiosonde observations used the geopotential height as the vertical coordinate.*

**Technical corrections:**

Page 7, eqn. (2):
*Ok, done.*

Page 7, eqn. (3):
and the three bracketed expressions need to be squared.
*Ok, done.*

Page 7, eqn. (4):

I suggest to replace the horizontal bars in eqn. (4) (rb;c and vb;c) by a more conventional notation indicating vectors

*Ok, done.*

Page 8, eqn. (6):
Here, in contrast to eqn. (4), the horizontal bar seems to differentiate between transmitter and receiver dipole vector. I suggest to clarify the notation.

*Ok, done.*

Page 13, line 10:
There appears to be a reference missing (empty bracket).

*Ok, was a typo left, corrected.*

**Response to Anonymous Referee #3's comments**

This paper introduces, in a comprehensive way, the data processing of the first Beidou based Chinese radio occultation mission FY-3C GNOS and 3-month data were used for the study/data processing. The two strategies of data processing investigated are zero-differencing and single-differencing. Differencing is a standard data process strategy in GNSS data process to mitigate (or cancel out) the various errors (e.g. signal generation/emission, signal propagation, signal transmission and signal reception) inherited with the technology. Various analyses of the atmospheric profiles based on the single- and zero-differencing data processing strategies and using three months' data, are carried out to evaluate the quality of BDS GNOS RO data and the robustness/quality of the zero-differencing data processing method. By comparing with ECMWF model and co-located radiosonde data, the BDS GNOS atmospheric profiles derived are fairly consistent.

Data processing algorithms are introduced in a fairly detailed way. The analyses are described and presented in a logical and clear manner. The discussions are comprehensive albeit some further clarification is needed. The conclusions given from the analysis are sound and reflect the current state-of-the-art in the field.

*Thank you.*

**Following are my other comments/suggestions for correction**

1) FY-3C GNOS receivers can receive both the GPS and BDS signals for navigation and occultation modules, therefore GNOS provides a different way to validate its BDS RO data (i.e. based on the zero-difference processing and GPS RO retrievals). I wonder the reason why not to use the GPS GNOS RO retrievals to validate BDS's counterparts?

*We agree that comparing with the GNOS GPS RO retrievals to validate BDS's counterparts is a good idea and a potential way to do the FY-3C GNOS RO data evaluation. However, the radiosonde observations and the ECMWF analysis data are reliable GNOS-independent data, which have previously been used as reference data to also validate GPS RO retrievals. Therefore, for this initial GNOS BDS evaluation we selected the radiosonde and ECMWF data as preferred source to use as reference to validate the BDS RO data. Nevertheless, since we could achieve a limited collocation ensemble of BDS RO and GPS RO, we included one BDS vs. GPS intercomparison figure now in section 3.2., which shows reasonably high consistency. Of course, further improvements and a detailed intercomparison analysis of the GPS and BDS RO data is a very interesting study, and we plan to do it by an extra paper.*

2) The current coverage of Beidou is regional. It would be great if the authors can comment over the issue of limited coverage of the Beidou system and how it affects the ROE occurrence?

*Thank you for pointing to this; we think, though, that in the view of the focus of this paper (an initial validation of the BDS RO profiles) we have commented on the current limitations of the BDS MEO, IGSO, and GEO subsystems in adequate length. We did so in the introduction, in section 3 where we also visualize the RO events occurrence in terms of the geographic coverage situation (Fig. 5), etc.*

3)
- Technical Corrections - Define the acronym for GRAS, GEO, IGSO and etc. when they appear in the first place in the text and use the acronyms thereafter.
*Ok, done.*

- Page 3, lines 13-14: the word "satellites" is repeated.
*Ok, corrected.*

- Page 13, line 10: It looks like you might be missing a reference here.
*Ok, a typo was left, corrected.*

- Page 16, lines 4-5: Should be Allan deviation (ADEV), not Allen variance.
*Ok, corrected.*

- Be careful with some reference formats and typos.
*Ok, looked again over the texts and further polished reference formats and typos.*

- be careful in using the differential technique, you need to be consistent to use differencing or differenced or difference. They do have minor differences. The "single-different" in figure 5 (a)/(b) is NOT right.
*Ok, corrected.*

- the title of the paper looks awkward and it needs to change "processing" and "data" need to be "data processing"
*Thank you, we carefully considered and tried this, but then preferred to keep the current formulation (expresses best in our view the aspect that we focus on the new BeiDou radio occultation data and that the key processing focus is excess phase processing). We made a little simplification, though, in leaving out the term "System" from the title, since "BeiDou radio occultation data" instead of "BeiDou System…" is sufficient in the title.*

- GNSS is commonly referred to The Global Navigation Satellite Systems (plural!!!)
*Yes, we agree this is done in particular if the plurality of the different constellations (in particular GPS, Glonass, BDS, Galileo) is emphasized. Since we use it as a generic term (collectively for the constellations), however, we generally prefer to follow the usual notation of using the term Global Navigation Satellite System as the name of the overall system.*

- the language usage needs to be sharpened and grammatical problems are spotted.
*Ok, as mentioned above, we rechecked all texts and the language usage has been improved.*

- "sub-global" needs to be replaced as "regional"
*Ok, done.*

**Response to Anonymous Referee #4's comments**

**General comments:**
The paper describes the evaluation of zero-difference processing vs. single-difference processing of BeiDou System (BDS) radio occultation (RO) data collected by the Chinese GNOS instrument on board the FY-C3 satellite.

Although this is not the first paper describing GNOS BDS retrievals, it is the first paper describing the zero- and single-differencing methods in that context, and comparing results using either method. For that reason it should be published in AMT. There are, however, a number of issues that needs to be addressed in a revised version.

The single- and zero-differencing methods are outlined and their application seems sound, although not all relativistic corrections seem to be adequately described. This, together with small unexpected differences in the results, gives me a grain of uncertainty as to whether the relativistic effects and clock offsets are correctly removed. I elaborate on this in one of the specific comments below.

*Thank you.*

Comparisons of derived bending angle and refractivity to reference profiles from ECMWF analyses are encouraging, although I do not think the results presented show that GNOS BDS RO data are of such high quality as claimed in the text. The authors mention (top of page 11) that part of the bias in their results could be from differences in vertical geo-location of GNOS and reference profiles (which are from ECMWF analyses and radiosondes). I'm not sure that such differences could give rise to the biases that are shown, but if so, such systematic difference should be better understood in the presented data set and possibly corrected. I am not aware of bias problems in the data evaluation of GPS RO data from other sources, e.g., COSMIC/CDAAC or Metop/EUMETSAT. In my comments to the results in Fig. 6 and 7 below, I point to a few other issues that are not mentioned in the text, but which should at least be discussed if improvement of the results in a revised manuscript is not possible.

If improvement is not possible, then some of the statements in the paper should be toned down, e.g., in the abstract where it says that "The statistical evaluation against these reference data shows that the results from single- and zero-difference processing are consistent in both bias and standard deviation, clearly demonstrating the feasibility of zero-differencing for GNOS BDS RO observations.", or at the end of the abstract where it says "The validation results establish that GNOS can provide, on top of GPS RO profiles, accurate and precise BDS RO profiles both from single- and zero-difference processing." Although the GNOS BDS RO data might be of a very high quality comparable to that of GPS RO data, such claims are not fully supported by the results in this paper.

*Thank you for these frank yet constructive comments. We agree that this paper is just an initial study to evaluate and show what we found to be a quite good quality already of the new GNOS BDS RO data. But of course, it clearly has further improvement potential from more rigorous future analysis, and we plan to do this by follow-on work. We therefore took your*

*suggestions serious, related to the scope and limitations of results of this initial paper, and carefully considered to tone down some statements in the paper; and we did so for several statements. For example, in the abstract we now say "are reasonably consistent" (instead of "are consistent") and "the validation results indicate" (instead of "the validation results establish"), etc.*

One thing that could give more confidence in the single- and zero-differencing results would be to show cases of ionospheric corrected excess phases at very high altitudes. Although the ionospheric correction in the processing is done at the bending angle level (at common impact parameters of B1 and B2 bending angles), excess phase data corrected at the same times could be shown for cases where the ionospheric residual is small (this could be based on the difference between B1 and B2 excess phases, choosing only cases where such difference/variation is small). If such cases, at altitudes above _100 km, show virtually no slope (giving confidence that the relativistic effects and clock offsets are correctly removed), and only noise at the level indicated in Fig. 3, then that would give added confidence in the quality of the data. A few examples together with statistical evidence that ionospheric corrected excess phases at high altitudes are virtually flat compared to the random noise, would make a very good case. Unfortunately, there is no method description of the derivation of bending angle and refractivity. Could such description be added (possibly just with reference to previous works)?

*We agree it is basically a very good suggestion to look into such ionosphere-corrected excess phases at high altitudes, complementary to the current upper troposphere/lower stratosphere (UTLS) validation and with more rigor. In this initial study on GNOS BDS RO evaluation we preferred to focus on UTLS validation against GNOS-independent reference data, however, as we started to do in previous papers (e.g., Liao et al. AMT 2016) for GNOS GPS RO evaluation. As noted in the first response above, we agree that this paper is an initial step only and that more rigorous inspection of small residual errors, of different possible sources within the orbit determination and excess phase processing, are needed and will be done by us in follow-on work.*

*For the retrieval of bending angle and refractivity profiles from the excess phase data, we used the ROPP software (available from the European ROM SAF consortium); we added a clarifying sentence to this end in section 3.2.*

A few additional questions comes to mind: Are there both setting and rising occultations in the statistics, and how many of each? How far down is the B2 signal typically tracked in rising and setting? How far up are the signals typically tracked? Are extrapolation of B1-B2 performed in the troposphere to extend profiles down to where B1 is tracked (if it is tracked lower than B2)?

*Response:* "Are there both setting and rising occultations in the statistics, and how many of each?" *Yes, the numbers of rising and setting are around half of the total number in Fig.1b, i.e., the contribution of setting and rising events to the total number is about the same.*

"How far down is the B2 signal typically tracked in rising and setting? How far up are the signals typically tracked?" *the B2 signal typically could track down to about 5 km, near half of them could track down to about 3 km, few of them could track down to 2 km or more.*

"Are extrapolation of B1-B2 performed in the troposphere to extend profiles down to where B1 is tracked (if it is tracked lower than B2)?" *We did not use the B1-B2 extrapolation in the BDS processing so far, but we consider to do it in future and will evaluate this further.*

Below I give specific comments and technical corrections with <page>/<line> referring to the pdf copy of the manuscript. In some places I give suggestions for improved language that could easy the readability, but not in all places where such improvement could be warranted. Suggested words are in square brackets. I kindly urge the authors to run the manuscript by a person with excellent skills in the English language.

**Specific comments and Technical corrections:**
1/3: Consider a small change to the title: "... data [from] the FY-3C GNOS mission"
*Ok, done.*

1/22: "[The] GNOS ..."
*Ok, done.*

1/26: "... on [the] FY-3C GNOS, [and] thus ..."
*Ok, done.*

2/12: Skip "as small as".
*Ok, done.*

2/13-14: Bad syntax: "including for the GEO, IGSO, and MEO subsets.". Could be skipped here, since you already indicated earlier in the abstract that the data are from these three sub-systems.
*Ok, done.*

2/14-15: "as may be expected from its lower vulnerability to noise." could also be skipped here.
*Thank you for the suggestion, we carefully considered it but we then preferred to keep this sentence here, to explain the potential reason.*

2/17: "... satellites [can] thus provide..."
*Ok, done.*

2/24-26: Move "Earth's" to before "atmospheric parameters...".
*Ok, done.*

3/9: "LEO" is not previously defined.
*Ok, done.*

3/20-23: I suggest reformulation, e.g.: "One of these LEO missions is the FengYun 3

series C satellite (FY-3C), carrying China's first GNSS Occultation Sounder (GNOS) (Liao et al., 2016). FY-3C was successfully launched on 23 September 2013."
*Thank you, done.*

3/28: I suggest reformulation, e.g.: "... satellites, the next being FY-3D, scheduled for launch in 2017, will also carry GNOS instruments, similar to ..."
*Thank you, done.*

4/1-3: Please reformulate. Are the antennas considered part of the instrument (line 1) or are they used by the instrument (lines 2-3)?
*Yes, all these antennas are used by the instrument.*

4/4: "... in [the] GNOS design.".
*Ok, done.*

4/5: "... from Earth's surface ..."
*Ok, done.*

4/13: Replace "it" with "the single-difference method".
*Ok, done.*

4/15: "... [the] single-difference ..."
*Ok, done.*

4/15-16: Redundant information (and bad syntax) that could be skipped: "during the GPS clock offset estimation process."
*Ok, done.*

4/17: "... needs no ground station data, [the] processing is simpler".
*Ok, done.*

4/24: "...requires that the LEO receiver [is equipped with] an ...".
*Ok, done.*

4/26: "... is [equipped with] such ...".
*Ok, done.*

4/31: "... received [the signals from five] geostationary ... (MEO) orbit satellites.".
*Ok, done.*

Section 1: Perhaps you could mention the B1 and B2 frequencies somewhere in the introduction. Section 1: Perhaps you could mention the different semi major axes and inclination of the GEO, IGSO, and MEO sub-systems somewhere in the introduction.
*Ok, done.*

5/23: "Recently, [because of] its higher complexity ...".
*Ok, done.*

6/1: "The inputs to [the processing] ...".
*Ok, done.*

6/20: "... for [the] receiver clock and [the] GNSS satellite clock ...".
*Ok, done.*

6/9-11: I do not understand this sentence: "Specifically, in this study, we use the BDS satellite data as orbital data inputs and outputs, while time-wise also using GPS time for the processing of the BDS data." Please clarify.
*It means, the GPS time is used for time-tagging also of the BDS observations, while the transmitter orbit data are of course the BDS data. We have clarified the sentence in the text.*

6/13: "(in units of [length]) at [carrier signal] i". (also in first line of page 7)
*Ok, done.*

6/16: Is there a reference for eq. (1)?
*Ok, done; we included Schreiner et al. (2010) now:*
*Schreiner, W., Rocken, C., Sokolovskiy, S., and Hunt, D.: Quality assessment of COSMIC/FORMOSAT-3 GPS radio occultation data derived from single- and double-difference atmospheric excess phase processing. GPS Solut., 14, 13-22, doi:10.1007/s10291-009-0132-5, 2010.*

6/17: Skip "(m/s)"
*Ok, done.*

7/4: Superscript on last term in eq. (2) should be "c".
*Ok, done.*

7/10: Superscript "a" should be "b" on the left-hand side of eq. (3).
*Ok, done.*

7/12: Shouldn't it be capital letters "B or C" here?
*Thank you, we chose to revise the labels 'A B C' to 'a b c' in Figure 1, to be more easily consistent everywhere.*

7/17: Please provide a reference for eq. (4). You say that this is a "periodic relativistic effect", but does not mention the main part of the relativistic correction, and it is therefore unclear if you make all the necessary corrections. If I understand relativistic effects in the GPS correctly, then eq. (4) is a residual that comes about because the GPS transmitters have their clocks

adjusted prior to launch, such that the GPS clocks in orbit beat at the same rate as a clock on the Earth (Ashby, Relativity in the Global Positioning System, Living Rev. Relativity, 6, (2003), 1, http://www.livingreviews.org/lrr-2003-1). However, part of that adjustment of the transmitter clock results in an additional frequency shift in ECI that must be taken into account in the zero-differencing (it cancel in the single-differencing). The shift is proportional to the effective gravitational potential at the surface of the rotating Earth, and is actually larger then the relativistic effect in orbit that would have been without this clock adjustment. See, e.g., eq. (46) in Ashby (2003). You should make clear how you make this additional correction. Ashby describes the relativistic effects in the GPS. Are clocks in the BDS similarly adjusted before launch? If so, it would be interesting if you could give the different values of the frequency adjustments in the BDS subsets (GEO, IGSO, and MEO). Also, eq. (4) is relevant for GNSS clocks (as you write), but what about the relativistic effects of the FY-3C satellite clock? They do not seem to be described? Nor is it mentioned how they are estimated in the zero-differencing. Again, eq. (46) in Ashby (2003) could be of help here. In any case, you should make clear how you estimate all the main relativistic effects and clock offsets (please also make clear whether you consider the correction for the transmitter clock adjustment part of the clock offset or part of the relativistic effects).

*Ok, done as good as we could for now. The reference for Eq. (4) is as well: Schreiner, W., Rocken, C., Sokolovskiy, S., and Hunt, D.: Quality assessment of COSMIC/FORMOSAT-3 GPS radio occultation data derived from single- and double-difference atmospheric excess phase processing. GPS Solut., 14, 13-22, doi:10.1007/s10291-009-0132-5, 2010.*

*And yes, the clocks of BDS are similarly adjusted prior to launch as for GPS. So the same equations as for GPS can be used for BDS data processing. In terms of the values of frequency adjustments, they depend on the orbit altitudes, BDS MEO satellite are set closely similar as GPS satellite; and the BDS GEO and IGSO satellite clocks are set to slightly different values. Currently in our data processing, we did not consider the LEO satellite relativistic effects but investigate in this direction for future updates.*

*We have included this type of explanations in the text below Eq. (4) now.*

7/18: Shouldn't there be bars above r and v here?
*Ok, done.*

7/18: Shouldn't it be "GNSS" instead of "GPS"?
*Ok, done.*

7/20: Please provide a reference for eq. (5).
*Ok done. It is again Schreiner et al (2010):*
*Schreiner, W., Rocken, C., Sokolovskiy, S., and Hunt, D.: Quality assessment of COSMIC/FORMOSAT-3 GPS radio occultation data derived from single- and double-difference atmospheric excess phase processing. GPS Solut., 14, 13-22, doi:10.1007/s10291-009-0132-5, 2010.*

7/20: Subscript "r" should be "a" five places in eq. (5).
*Ok, done.*

8/2: I think eq. (6) needs to be multiplied by the i'th wavelength to be consistent with the terms in eq. (1) and (2).
*Ok, done.*

8/3: In eq. (4) bars were used to indicate vectors, so it is unfortunate here to distinguish the two effective dipole vectors by a bar on one, but not the other. I suggest to use another distinction and consistently use bars to indicate vectors.
*Ok, done. We have used the bold italic letters as vectors.*

Section 2.1: Generally, It would make good sense to mention the order of magnitude of different corrections and their relative importance.
*Now that we have the references included we considered it makes the text less concise to read if we include also this. Also for the purpose of this paper it is an introductory description to the excess phase equations; in the follow-on study looking in detail into the improvement of remaining small residual errors in the excess phase processing we would of course intend to describe these aspects in more detail.*

8/6: I suggest skipping "adopt".
*Ok, done.*

8/10: I suggest to remove "employing Eq. (2),".
*Ok, done.*

8/14-16: Some of the "a" and "c" subscripts and superscripts on the right-hand sides of eq. (7) and (8) should be interchanged.
*Ok, done.*

8/14-16: I suggest the use of different symbols in eq. (7) and (8) (and similar in eq. 9) for the phases on the right-hand side, since these are corrected for the effects mentioned in the first paragraph of this section. Perhaps you could simply use a tilde to indicate that they are not strictly the same as the ones in eq. (1) and (2), and at the end of the first paragraph in section 2.2 (line 9) you could write something like: "In the following we refer to these as <symbol_ab> and <symbol_ac>, respectively."
*Ok, done.*

8/18: I suggest replacing ". c1 and c2 are just" with "are".
*Ok, done.*

9/18: You could say "mentioned" instead of "aforementioned".
*Ok, done.*

9/21: Could the occulting and reference satellites be from two different sub-systems, e.g., a MEO as reference for an occulting GEO?

*Yes.*

10/1-2: It is not clear how you calculate the "carrier phase observation error standard deviation" shown in Fig. 3. Are you applying a high-pass filter? With what band-width?

*Yes, in the processing a high-pass filter has been used and the band-width chosen was seven seconds. Please refer to: Oliver Montenbruck, Yago Andres, Heike Bock, et al. (2008); Tracking and orbit determination performance of the GRAS instrument on MetOp-A. GPS Solut., 289-299.*

11/24 (and other places): You could use the word "difference" instead of "error".

*We considered this, but given the rest of the relevant notation as used in this paper, we preferred to keep the current terminology also here.*

11/25: Use a mathematical symbol for bending angle in eq. (11) (BA is an abbreviation, not a symbol).

*Ok, done, we use the greek symbol Alpha now, which is often used for bending angle in the RO community.*

11/28: You could here introduce the use of "Bias" and "StdDev" as they are used later in the text: "... estimates of biases (Bias) and standard deviations (StdDev) are illustrated ...".

*Ok, done.*

12/27-28: I do not understand the sentence in parenthesis: "though more standard deviation suppression might be expected from avoiding the reference link computation". It is not clear what "standard deviation suppression" mean, and I'm not sure if this statement is different from what you just said in the sentence before? Using the word "though" indicates that it is contradicting what you said before. Please clarify.

*Ok, clarified.*

12/28: Schreiner et al. 2009 is not in the reference list. Should perhaps be 2010.

*Ok, done.*

13/3: "Scherllin" instead of "Scherrlin".

*Ok, done.*

13/10: Empty parenthesis. Perhaps a reference is missing.

*Ok, a typo was left, corrected.*

13/11-12: Is it really the first time that RO retrievals from other than the BDS MEO is demonstrated? Liao et al. (2016) also describes the GNOS-BDS occultation coverage using BDS GEO and IGSO, and I could not find any indication in their paper that the statistics they show is only from MEO occultations. If it is the first time, then you should here make clear that the results in Liao et al. (2016) did not include GEO and IGSO occultations.

*Ok, revised.*

14/3: You say that GNOS BDS retrievals are comparable to GPS retrievals, but you have not really shown that here. Either you should show comparisons to GPS retrievals, or you need to support such statements with citations to previous works.
*We agree that comparing with the GNOS GPS RO retrievals to validate BDS RO retrievals is a good idea and a potential complementary way to do the FY-3C GNOS RO data evaluation. However, the radiosonde observations and the ECMWF analysis data are reliable GNOS-independent data, which have previously been used as reference data to also validate GPS RO retrievals. Therefore, for this initial GNOS BDS evaluation we selected the radiosonde and ECMWF data as preferred source to use as reference to validate the BDS RO. Nevertheless, since we could achieve a limited collocation ensemble of BDS RO and GPS RO, we included one BDS vs. GPS intercomparison figure now (in section 3.2.), which shows reasonably high consistency. Of course, further improvements and a detailed intercomparison analysis of the GPS and BDS RO data is a very worthwhile next study as well, and we plan to do it by an extra paper.*

14/5: "... not [only] on MEO satellites but [also] on GEO and IGSO satellites.".
*Ok, done.*

14/26: You say that "Single-differencing does not need to correct the receiver clock offset". I know what you mean, but it is not strictly correct. The receiver clock offset is removed because it cancel in the single-differencing. Please reformulate.
*Thank you, done.*

15/2-4: You say that in the zero-differencing there can be some residual errors after the clock offset correction, but you have not shown that anywhere. Can you give examples of such residuals?
*After the LEO clock correction in zero-difference, the LEO clock left a residual errors, which could be estimated by stability of the LEO clock. We can use the Allan deviation to describe it; for more information you can refer to Figure 10 in*
*"Cai Y, Bai W, Wang X, et al. In-Orbit Performance of GNOS on-board FY3-C and the Enhancements for FY3-D Satellite. Advances in Space Research, 2017."*

16/4: It should be "Allan", not "Allen".
*Ok, done.*

16/5: Please reformulate the statement on the Allan variance (or deviation) here. It is correctly formulated in the abstract. The unit is not second.
*Ok, done.*

16/10-12: The last paragraph should be reformulated or removed. It is unclear what "in this context of the leading instruments" means.
*Ok, done.*

Results shown in Figure 6:

1) StdDev: I would have expected visible differences between single- and zerodifferencing to be only at high altitudes/impact heights. However, even below 10 km it seems that the StdDevs are significantly different, with the zero-differencing results generally having the larger StdDev. How can that be explained? From the legend it appears that it is exactly the same number of occultations involved (and I assume therefore that it is the same occultations). Also, it seems that the StdDev starts increasing already at 20-25 km. I would have expected the increase to start a bit higher when I compare with GPS RO statistics from other sources.

*For the StdDev profile differences below about 10 km and above about 20-25 km, the reasons are likely somewhat complicated (several possible sources in the excess phase processing, also BDS ephemeris, esp. for GEO) and we are analyzing this type of differences in follow-on work to this initial study. We have included a clarifying sentence explicitly pointing to this.*
*On the "same number of occultations", yes, both ZD and SD involve the same RO events.*

2) Bias: The differences in bias between single- and zero-differencing are similar below 20 km, and the somewhat larger negative bias for GEO can probably be explained by the fact that all the GEO occultations are at very high altitudes (and for some reason that gives a larger negative bias when compared to ECMWF). However, above 20 km, the biases for the three subsystems (MEO, IGSO, GEO) are diverging more for the zero-differencing results, and in particular the bias for the GEO occultations becomes more negative than it is for single-differencing. Why?
These issues needs to be discussed in the text.

*See the previous response above, which in general also holds here; more detailed follow-on work will clarify the more subtle differences. We have include an additional sentence related to the specifics of the GEO RO events now as well, pointing to the fact that the GEO orbit determination is the most challenging from all BDS and that the GEO RO events are restricted to high latitudes only (as visible in Fig. 5), i.e., a potential regional selection effect.*

Results shown in Figure 7:

The same comments as above applies here, but additionally, it is very strange to see the bias for the GEO occultations for single-diffencing at high altitudes being more positive than the others. This is inconsistent with the biases in the bending angle. It is critically important to understand this, since you are trying to make the point that zero-differencing has lower StdDev than single-differencing, but it is difficult to have confidence in the results if there are such inconsistencies in the biases.

*On top of what is said above related to the bending angle Figure, we also included a sentence here for refractivity, pointing to the follow-on work for detailed error analysis and to the specifics of the GEO results.*

Figure 8 axes labels: I suggest to redo this figure with labels as in Figure 7 ("R%" does not make sense; "geop" should be "Geopotential height").
*Ok, done.*

[revised manuscript text omitted]